# LegoMT2: Non-Blocking Federated Learning for Massive Multilingual Machine Translation

## Abstract

What is the maximal number of languages that a single machine translation model can translate? It is a critical challenge to learn a single model for massive languages. Prior methods focus on increasing the model size and training data size. However, large models are difficult to optimize efficiently even with distributed parallel training and translation capacity can interfere among languages. To address the challenge, we propose LegoMT2, an efficient approach with a tailored model architecture for massive multilingual neural machine translation. LegoMT2 organizes 435 languages into 8 language-centric groups and attributes one local encoder-decoder for each group and a global encoder-decoder for all languages. LegoMT2 then trains each local and global encoder-decoder on a group-dedicated set of clients through asynchronous updating of parameters. We trained LegoMT2 on a large dataset with 25 billion sentence pairs beyond English-centric. LegoMT2 is $16.2\times$ faster than the distributed training method for the same-size NLLB while improving the translation results by an average of 2.2 BLEU on *Flores-101* [1].

## 1 Introduction

Recent years have witnessed great success in multilingual neural machine translation (MNMT) (Ha et al., 2016; Johnson et al., 2017; Bapna et al., 2019; Liu et al., 2020; Fan et al., 2021; Costa-jussà et al., 2022) that uses a single model for translating all directions. To construct an MNMT system that supports high-quality translation for massive directions, many efforts have been put into scaling up the model size and training corpus (Liu et al., 2020; Fan et al., 2021). For example, Costa-jussà et al. (2022) constructed a 54.5B NLLB model to support translation among 200 languages. Additionally, recent advancements in large language models, such as GPT-4 (OpenAI, 2023) and LLAMA (Touvron et al., 2023), have shown promising potential in multilingual machine translation. Generally, these multilingual models are also trained using a single model.

However, with the increasing model size, training a single model over massive data brings new challenges. Specifically, the challenge is two-fold: (1) *huge training costs*. Training and serving a large MNMT model requires a pile of GPUs associated with massive communication costs for aggregation among different devices (Johnson et al., 2017; Fan et al., 2021), which brings huge training delays and thus largely reduces training efficiency (Rasley et al., 2020; Narayanan et al., 2021); (2) *parameter interference*. Parameter interference is a fundamental problem in multilingual machine translation. It refers to the competition between different languages for the limited parameters of a model when we hope to use a single model to handle all translation directions. This can result in good translation results for some languages, while the translation results for other languages may be less satisfactory. Previous studies have observed parameter interference, especially when dealing with numerous translation directions (Aharoni et al., 2019; Gordon et al., 2021; Yang et al., 2022; Fan et al., 2021). Test error often falls off as a power law with model size in machine translation. Mixture-of-Experts (MoE) (Jacobs et al., 1991; Shazeer et al., 2017; Lepikhin et al., 2020; Fedus et al., 2021; Du et al., 2022; Fan et al., 2021; Costa-jussà et al., 2022) is a popular solution to reduce parameter interference, but it also introduces substantial memory and computational requirements.

---

[1]We will release the model and code to the public.

To address these challenges, we propose LegoMT2, an efficient approach to massive MNMT. LegoMT2 consists of three key designs: a proper language grouping scheme, a tailored multi-way model architecture, and a non-blocking federated learning algorithm.

First, LegoMT2 splits data into carefully designed language groups. This grouping affects our method design and training algorithm. Under this scheme, we arrange all languages based on the size of the language-centric data (sentence pairs that are from or to a specific language) and divide this language-centric data into 8 different groups of equal size. Each group may contain a different numbers of languages. Each group's data is stored on a dedicated set of GPU servers, therefore no moving of training data is needed.

Second, we design a multi-way detachable model to alleviate parameter interference. Our key insight is the separation of the model used for training and inference and splitting language capacity into different model components. The model at training time includes but can be much larger than the inference model. Our multi-way model consists of a global encoder-decoder for all languages and one local encoder-decoder for each language group. In total, the model has 9 encoder-decoders at training time. At inference time, it only uses the global encoder-decoder. The model architecture also affects our algorithm design decisions.

Third, we design a non-blocking distributed learning algorithm to accelerate the training. Our key insight is that at training time, we no longer need to load all model parameters (for 9 encoder-decoder) into all servers, thanks to our language grouping scheme and associated multi-way architecture. We dedicate one set of servers to one language group. We only load and train the encoder-decoder parameters responsible for the group, plus the global encoder-decoder. A separate thread is responsible for aggregating the global parameters across different servers. Parameter communication is asynchronous and efficient, which does not block the training on local servers. We only need to transfer the global parameters from servers at intervals. The need for transferring local encoder-decoder is eliminated, thereby reducing communication costs. While asynchronous training has been studied before, our work is the first to demonstrate its effectiveness in massive MNMT training.

We construct a large-scale MNMT translation dataset to train LegoMT2. The proposed dataset contains 25B parallel pairs, covering 435 languages and 22,613 translation directions. Our contribution can be summarized below:

- We propose an efficient training framework LegoMT2 for MNMT. LegoMT2 is empowered by an efficient non-blocking optimization algorithm to accelerate training and a tailored multi-way detachable model architecture.
- We design a proper grouping scheme of 435 languages and 22k language directions. Our approach properly attributes one encoder-decoder to each group, with which we train a 1.6B LegoMT2 model for 435 languages.
- Our experiments on *Flores-101* show that LegoMT2 achieves 16.2× speedups and 2.2 BLEU gains over the prior best approach.

## 2 RELATED WORK

The most common approach in MNMT is using a single model to handle all translation directions Ha et al. (2016); Johnson et al. (2017); Bapna et al. (2019); Liu et al. (2020); Fan et al. (2021), which has promising generalization abilities by transferring knowledge from high-resources and low-resources. Nevertheless, researchers Aharoni et al. (2019) have observed that there is the trade-off between translation quality and language number when using a single model for inference. Federated learning is originally proposed to address privacy problems. McMahan et al. (2017) first introduced *Federated Learning* and applied this algorithm in both computer vision and NLP tasks. Since then, more and more studies have explored NLP models with federated learning (Sui et al., 2020; Lin et al., 2021; Passban et al., 2022; Weller et al., 2022; Tian et al., 2022). Sui et al. (2020) focused on efficient federated communication methods. Lin et al. (2021) evaluated different federated methods on various NLP tasks. Passban et al. (2022) introduced federated learning into multi-domain translation. Most of them focused on using federated learning to better fine-tune a pre-trained model. Unlike these studies, Tian et al. (2022) proposed a framework that collaboratively pre-trained a BERT model with privacy data in a federated way. In this paper, we propose a new training recipe for MNMT pre-training based on the Federated Learning framework.

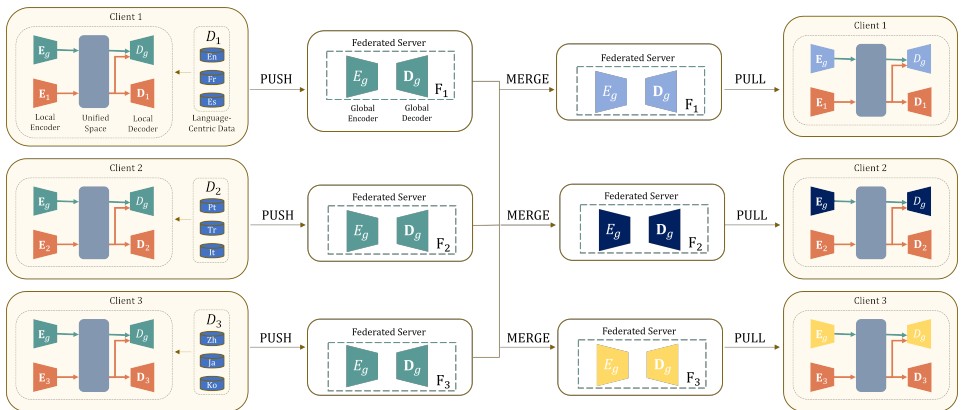

Figure 1: Overview of LegoMT2. It partitions data into language-centric groups. E.g. all parallel sentences from/to English, French, and Spanish are in Group 1 and stored on client 1. The model consists of a global encoder-decoder for all languages and multiple local encoder-decoder for specific language groups. During training, each client transmits its global encoder-decoder parameters to the federated server (PUSH) and gets the updated global parameters from the server (PULL) at pre-determined intervals. Parameter communication is asynchronous, which does not block the training on local clients.

## 3    THE LEGOMT2 APPROACH

### 3.1    OVERVIEW

Our goal is to develop a single model to translate massive languages (over 400). Prior approach needs to scale the model to extremely large (e.g. 54 billion parameters for NLLB), which is costly to train. We aim to tackle this challenge through a holistic approach considering three aspects: a proper language/data grouping scheme, a tailored architecture, and a more efficient distributed training algorithm. We design LegoMT2 approach with tailored components in all three aspects (Figure 1).

First, LegoMT2 includes a group scheme that arranges all sentence pairs from and to a specific language into a group. Our goal is to balance the number of parallel sentences in each group so that each contains equal data size. Figure 1 shows 3 groups while in our experiment we use 8 groups.

Second, LegoMT2 uses a multi-way model that includes multiple encoder-decoders with shared embedding space. LegoMT2 includes one local encoder-decoder for each language group and one global encoder-decoder for all languages. The purpose is to alleviate parameter interference while keeping the multilingual capability. As shown in Figure 1, by incorporating global encoder-decoder ($E_g - D_g$), LegoMT2 ensures the sharing of essential knowledge across all language groups, facilitating the accumulation of collective intelligence during the training process. Simultaneously, the local encoder-decoder, such as $E_1 - D_1$, $E_2 - D_2$ or $E_3 - D_3$, allow for fine-tuning the adaptation to address the unique characteristics and challenges of individual languages.

Third, LegoMT2 provides a non-blocking training algorithm, as illustrated in Figure 1. Each server stores a local and a global encoder-decoder. It calculates gradient updates for two encoder-decoders using the language group data stored on the server. Each server only pushes the global encoder-decoder parameter to the central server and pulls from that at pre-defined intervals. This update is asynchronous and minimizes the parameters for transferring across servers.

### 3.2    LANGUAGE GROUPING SCHEME

We assign different data groups to different clients. Given a multilingual parallel dataset $\mathcal{D}$ with $N$ languages, $\mathcal{D} = \{\mathcal{D}_{1\rightarrow\cdot}, \mathcal{D}_{\cdot\rightarrow 1}, \cdots, \mathcal{D}_{n\rightarrow\cdot}, \mathcal{D}_{\cdot\rightarrow n}, \cdots, \mathcal{D}_{N\rightarrow\cdot}, \mathcal{D}_{\cdot\rightarrow N}\}$, where $\mathcal{D}_{n\rightarrow\cdot}$ refers to a parallel data from the $n$-th source language to any language except itself. $\mathcal{D}_{\cdot\rightarrow n}$ refers to a parallel data from other languages to the $n$-th language. The combination of $\mathcal{D}_{\cdot\rightarrow n}$ and $\mathcal{D}_{n\rightarrow\cdot}$ is language-centric data. Then we split all $N$ language-centric data into $P$ clients, that is, reorganizing $\mathcal{D}$ into

$\{\mathcal{D}_{S_1}, \cdots, \mathcal{D}_{S_P}\}$. The non-identical distribution for clients $i$ and $j$ is $\mathcal{D}_{S_i} \neq \mathcal{D}_{S_j}$.

$$|S_1| + \cdots + |S_i| + \cdots + |S_P| = N, \ S_i \cap S_j = \varnothing, \ |S_i| \neq 0 \tag{1}$$

where $S_i$ is a language set contains one or many languages; $|S_i|$ refers to the number of languages. $\mathcal{D}_{S_j}$ is the combination of language-centric data covering all languages in $S_j$.

For instance, a dataset $\mathcal{D} = \{\mathcal{D}_{\text{En}\rightarrow\text{Fr}}, \mathcal{D}_{\text{Fr}\rightarrow\text{En}}, \mathcal{D}_{\text{Zh}\rightarrow\text{Nl}}, \mathcal{D}_{\text{Nl}\rightarrow\text{Zh}}, \mathcal{D}_{\text{Fr}\rightarrow\text{Nl}}, \mathcal{D}_{\text{Nl}\rightarrow\text{Fr}}\}$ with $N = 4$ languages needs to split into 3 different clients with $S_1 = \{\text{En, Fr}\}$, $S_2 = \{\text{Zh}\}$ and $S_3 = \{\text{Nl}\}$ language sets. The result is $\mathcal{D}_{S_1} = \{\mathcal{D}_{\text{En}\rightarrow\text{Fr}}, \mathcal{D}_{\text{Fr}\rightarrow\text{En}}, \mathcal{D}_{\text{Fr}\rightarrow\text{Nl}}, \mathcal{D}_{\text{Nl}\rightarrow\text{Fr}}\}$, $\mathcal{D}_{S_2} = \{\mathcal{D}_{\text{Zh}\rightarrow\text{Nl}}, \mathcal{D}_{\text{Nl}\rightarrow\text{Zh}}\}$, $\mathcal{D}_{S_3} = \{\mathcal{D}_{\text{Fr}\rightarrow\text{Nl}}, \mathcal{D}_{\text{Nl}\rightarrow\text{Fr}}, \mathcal{D}_{\text{Zh}\rightarrow\text{Nl}}, \mathcal{D}_{\text{Nl}\rightarrow\text{Zh}}\}$. Here, we arrange languages based on the size of the language-centric data and divide them into different groups of equal size. Further details in our implementation are described in Appendix C.

## 3.3 Multi-way Model Architecture

LegoMT2 is not constrained to a specific implementation of the backbone model (e.g, Shared mode +Adapter (Houlsby et al., 2019), multi-way model (Fan et al., 2021; Yuan et al., 2022)). For simplification, we adopt a multi-way detachable model Yuan et al. (2022) with standard Transformer architecture, which decomposes the MNMT model with a global encoder-decoder and a local encoder-decoder. To clarify, each client possesses a local language-specific encoder-decoder and a duplicate of the global encoder-decoder. This setup can also be applied to structures that only contain a decoder. It's important to note that only the global module is shared across all devices. Although the local module is not shared, its parameters are subject to adjustment through the shared global module.

LegoMT2 is also not limited to a specific initialization. To minimize training costs, we utilize NLLB-200-1.3B to initialize both global and local parameters. To accommodate a large number of languages, we expanded the size of the vocabulary from 256K to 490K tokens. This is achieved by training Byte Pair Encoding (BPE) separately for each language and then merging these vocabularies. We use pre-trained embeddings for the tokens that are already in the original vocabulary and randomly initialized embeddings for the new ones.

To train the global and local parameters, we follow the three data flows in Yuan et al. (2022) to train client parameters: a global encoder with a local decoder (Dec-Flow), a global encoder with a global decoder (Mix-Flow), and a local encoder with a global decoder (Enc-Flow). Each flow can be used independently during the inference phase.

**Mix-Flow** Mix-Flow uses a global encoder and global decoder. The loss for it of client $i$:

$$F_{i_M} = \sum_{\mathbf{x},\mathbf{y}\sim\mathcal{D}_{i_{\text{multi}}}} -\log P_{\theta_m^i}(\mathbf{y}|\mathbf{x}) \tag{2}$$

where $(\mathbf{x}, \mathbf{y})$ is a sample from multilingual data, the parameters of Mix-Flow are $\theta_m$ and the probabilities output by the decoder is $P_{\theta_m^i}$. The multilingual data, including one-to-many dataset ($\mathcal{D}_{S_i\rightarrow\cdot}$) and many-to-one dataset ($\mathcal{D}_{\cdot\rightarrow S_i}$), for client $i$ is denoted as $\mathcal{D}_{i_{\text{multi}}} = \mathcal{D}_{\cdot\rightarrow S_i} \cup \mathcal{D}_{S_i\rightarrow\cdot}$.

**Enc-Flow** The loss for the Enc-Flow, which employs a local encoder and a global decoder:

$$F_{i_E} = \sum_{\mathbf{x},\mathbf{y}\sim\mathcal{D}_{S_i\rightarrow\cdot}} -\log P_{\theta_e^i}(\mathbf{y}|\mathbf{x}) \tag{3}$$

where $(\mathbf{x}, \mathbf{y})$ is a sample from one-to-many ($\mathcal{D}_{S_i\rightarrow\cdot}$) training data, $P_{\theta_e}$ is probability output by the decoder of Mix-Flow, $\theta_e$ is the parameters of Enc-Flow, and $i$ is client id.

**Dec-Flow** Dec-Flow uses a global encoder and a local decoder. The loss for client $i$ is:

$$F_{i_D} = \sum_{\mathbf{x},\mathbf{y}\sim\mathcal{D}_{\cdot\rightarrow S_i}} -\log P_{\theta_d^i}(\mathbf{y}|\mathbf{x}) \tag{4}$$

where $(\mathbf{x}, \mathbf{y})$ is a sample from many-to-one ($\mathcal{D}_{\cdot\rightarrow S_i}$) data, $P_{\theta_d}$ is the parameters of Dec-Flow.

The training strategy follows the approach outlined by Yuan et al. (2022) for individual clients $i$:

**Stage 1:** Training $f_i = F_{iM} + F_{iE}$ on client $i$ where the multilingual encoder/decoder and local encoder are trained together. This stage necessitates collaboration among all clients.

**Stage 2:** After stage 1 is completed, fixing the multilingual decoder and utilizing $F_{iD}$ to train the local decoder. This step is executed independently on a single client.

As observed, the performance of the global decoder is directly affected by the local encoders. However, the local decoder has no impact on the global module. Research conducted by Yuan et al. (2022) demonstrates employing local decoders potentially causes a significant shift in the distribution of multilingual encoders, leading to catastrophic forgetting.

Formally, the training objective for the whole system is $F(\theta) = \mathbb{E}_{i \sim P}[F_i(\theta)]$, where $F_i(\theta) = \mathbb{E}_{x \sim \mathcal{D}_i}[f_i(\theta, x)]$, $\theta$ refers to the parameter of the target model; $F_i$ represents the local objective function at client $i$; $P$ is the total number of client; $\mathcal{D}_i$ is the data distribution in client $i$ and $f_i = F_{iM} + F_{iE}$. The function $F_{i_D}$ is used on the client $i$ and does not impact the overall system.

### 3.4 NON-BLOCKING OPTIMIZATION ALGORITHM

Large-scale training usually requires massive communication costs to collect gradients from each client. LegoMT2 develops an effective communication approach by exchanging parameters in an asynchronous way to broadcast global parameters across different clients and a federated server. In this work, the file system serves as the actual server, and the number of requests from LegoMT2 is significantly lower than the file system's maximum load capacity. The whole non-blocking federated learning comprises three main operations: *PUSH*, *MERGE*, *PULL*, as shown in Figure 1.

*PUSH:* In contrast to traditional federated learning approaches, where the shared module is uploaded to the server only after local training, LegoMT2 operates differently. It employs an asynchronous approach by saving the global module to the federated server at regular intervals $\alpha$ during training.

*MERGE:* In traditional federated learning, the server is required to wait until it collects the global encoder-decoder from all clients before merging them to generate a unified global model. In LegoMT2, the server can directly merge (simply average) global models that have been pushed to the federated server without waiting for the arrival of all models from all clients.

*PULL:* Each client will pull the latest fusion model from the federated server to update (the newest version overwrites the existing one) its local server every fixed interval $\beta$.

LegoMT2 uses these three operations to complete parameter communication across all clients. These functions do not pause the training of local clients, therefore largely improving the throughput of models. The whole training algorithm is shown in Algorithm 1.

## 4 EXPERIMENTS

### 4.1 DATASET, MODELS AND TRAINING DETAILS

**Training Set:** We gather many-to-many dataset from OPUS, an open corpus that compiles numerous parallel sentences from the internet, covering a wide range of domains, from legislative to religious texts. The dataset we constructed consists of 435 languages and approximately 22,000 language pairs, comprising around 25 billion sentence pairs. In the training set, over 11,000 language pairs contain more than 1,000 sentence pairs, and 1,151 of them have more than 1 million sentence pairs. Among all the languages, 19 have more than 1 billion sentence pairs (see more in Appendix B).

**Metric:** To evaluate the effectiveness of our model, we have taken a comprehensive approach. Since no dataset currently covers 400 languages, we have partially followed the standard testing process and assessed our model's performance on the widely-used multilingual dataset known as *Flores-101*. We use the same evaluation metric of sentence piece BLEU (abbreviated as **spBLEU**) to compare our approach with strong baselines and present the average performance of the 86 languages[2] that overlap with *Flores-101* for all M2M-100 models. Additionally, there is no parallel evaluation data for the majority low-resource languages that are not in *Flores-101*. we have employed back translation (*src-tgt-srcb*) to evaluate our model's performance over 435 language translations. This process involves translating text from the source language (*src*) to the target language (*tgt*) and then back to the source language (*srcb*). **Back-spBLEU** evaluates the spBLEU score between *src* and *srcb*. To avoid counting direct copies, we also report the translation performance between *src* and *tgt*.

---

[2]These 86 languages are: af, am, ar, ast, be, bg, bn, bs, ca, ceb, cs, cy, da, de, el, en, es, et, fa, ff, fi, fr, ga, gl, gu, ha, he, hi, hr, hu, hy, id, ig, is, it, ja, jv, ka, kk, km, kn, ko, lb, lg, ln, lo, lt, lv, mk, ml, mn, mr, ms, my, ne, nl, no, ns, oc, or, pa, pl, ps, pt, ro, ru, sd, sk, sl, so, sr, sv, sw, ta, th, tl, tr, uk, ur, uz, vi, wo, xh, yo, zh, zu.

---

**Algorithm 1:** Non-Blocking Federated Training

---

**Data:** Given $P$ clients and client-centric data, with predefined values for $\alpha$ and $\beta$ such that $\alpha < \beta$. Here, $\alpha$ represents the frequency in minutes at which the latest model is pushed to the federated server, while $\beta$ represents the frequency in minutes at which the latest model is pulled from the federated server.

**for** *client $i = 1$ to $P$* **do**
    Shuffle client-centric data to obtain a new client-centric training sequence $\mathcal{B}$ ;
    Record the save start time as $t_s$ and the load start time as $t_l$ ;
    **for** *batch $b = 1$ to $\mathcal{B}$* **do**
        Record the current time as $t_c$ ;
        **if** $t_c - t_s \geqslant \beta$ **then**
            $\theta_{\mathrm{avg}}^i \leftarrow \mathrm{MERGE}\,(\{\theta_m^1, \theta_m^2, \cdots, \theta_m^P\})$;     // running on the central server
            $\theta_m^i \leftarrow \mathrm{PULL}\,(\theta_{\mathrm{avg}}^i)$ ;     // running on the client $i$
            $t_s \leftarrow t_c$ ;     // running on the client $i$
        **end**
        **if** $t_c - t_l \geqslant \alpha$ **then**
            $\mathrm{PUSH}\,(\theta_m^i)$ ;     // running on the client $i$
            $t_l \leftarrow t_c$ ;     // running on the client $i$
        **end**
    **end**
**end**

---

**Models: Flores-175MB / 615MB** are two baselines released with the *Flores-101* dataset (Goyal et al., 2022), which are based on M2M-100 model. **M2M-100-1.2B** (Fan et al., 2021) is a powerful multilingual sequence-to-sequence model that can translate between 100 languages in 9,900 directions. It is an encoder-decoder model trained for Many-to-Many multilingual translation and built using the Transformer architecture. **M2M-100-12B** (Fan et al., 2021) is a multilingual encoder-decoder (seq-to-seq) model that builds on M2M-100-1.2B by adding language-specific information. Its main purpose is to perform translation tasks between any of the 100 languages. **NLLB-200-1.3B** (Costa-jussà et al., 2022) is a distilled variant of the NLLB-200 model, which is a pre-trained MNMT model that supports 200 languages.**NLLB-200-54.5B** (Costa-jussà et al., 2022) is a Mixture of Experts (MoE) model and is the largest MT model. To ensure a fair comparison, we fine-tune the NLLB-200-1.3B model on our datasets using a standard centralized training method, recorded as **Single-FT**.

**LegoMT2 Parameters:** We use a Transformer with 24 encoder-decoders. Given a vocabulary size of 490k and an embedding dimension of 1024, the total number of parameters for the embedding amounts to 0.5 billion, record as $\#\mathrm{embedding} = 0.5\mathrm{B}$. The embedding weight is shared between all encoder-decoder. A single encoder-decoder, comprising 24 transformer encoder layers and 24 transformer decoder layers, has a total parameter count of 1.1 billion, recorded as $\#\mathrm{encoder} - \mathrm{decoder} = 1.1\mathrm{B}$. During training, the total number of parameters of LegoMT2 is: $\#\mathrm{embedding} + \#\mathrm{encoder} - \mathrm{decoder} \times 9 = 0.5 + 1.1 \times 9 = 10.4\mathrm{B}$. During inference, we only use the multilingual global encoder and the multilingual global decoder, therefore the total number of parameters is $\#\mathrm{embedding} + \#\mathrm{encoder} - \mathrm{decoder} = 0.5 + 1.1 = 1.6\mathrm{B}$.

**Training Details:** We split training into 8 language groups in our framework. For balanced training, we sort all languages based on language-centric data and uniformly split all languages into 8 groups. The language details of 8 groups can be found in Appendix C. The training code is developed on fairseq[3] repository. The model architecture follows the design in Yuan et al. (2022), with different configurations and vocabulary size. Both the global and private models are initialized with NLLB-200-1.3B weights. In order to synchronize the speed among different clients as much as possible, GPU resources are allocated to each group as follows: each client model is trained on 8 80G A100-chips using the Adam optimizer with $\beta_1 = 0.9$, $\beta_2 = 0.999$, learning rate $1e - 4$, the maximum number of tokens in a batch is $4,000$, update parameters every $48$ batch, when in an epoch. The interval of save $\alpha$ and load $\beta$ is set as $6$ and $12$, respectively. This setting primarily considers the fault tolerance time for three consecutive loading failures.

## 4.2 EXPERIMENTAL RESULTS

**LegoMT2 outperforms single-model fine-tuning by a large margin.** As illustrated in Table 1, LegoMT2 outperforms *Single-FT* by a large marge with 2.2 spBLEU on many-to-one translation and

---

[3]https://github.com/facebookresearch/fairseq

Table 1: Result on the *Flores-101* devtest. "Para." refers to the number of parameters required for inference. "H" and "L" represent average results from or to high/low-resource languages, where high-resource languages include all languages in Families 1-6 while low-resource languages include all languages in Families 7-8. Single-FT and LegoMT2 have the same training data and can be fairly compared. LegoMT2, supporting 435 languages, outperforms Single-FT by a large margin.

| Model | H X→En | L X→En | H X→Pt | L X→Pt | H X→Hu | L X→Hu | H X→Da | L X→Da | H X→Zh | L X→Zh | H X→Sw | L X→Sw | H X→Pa | L X→Pa | AVG. |
|---|---|---|---|---|---|---|---|---|---|---|---|---|---|---|---|
| NLLB-200-54.5B | 44.9 | 39.0 | 35.8 | 30.8 | 27.8 | 22.8 | 34.6 | 28.7 | 17.3 | 16.7 | 28.4 | 25.4 | 30.7 | 27.0 | 29.3 |
| Flores-175M | 23.5 | 8.4 | 23.5 | 7.8 | 15.8 | 5.3 | 20.9 | 5.4 | 10.7 | 3.6 | 12.3 | 4.5 | 2.3 | 1.3 | 10.4 |
| Flores-615M | 30.9 | 12.8 | 30.1 | 11.8 | 22.0 | 8.0 | 27.5 | 9.6 | 15.9 | 6.2 | 18.6 | 7.4 | 3.7 | 2.1 | 14.8 |
| M2M-100-1.2B | 36.3 | 16.8 | 33.1 | 14.8 | 24.8 | 10.4 | 31.0 | 13.0 | 18.3 | 7.8 | 20.6 | 9.7 | 3.7 | 2.5 | 17.3 |
| M2M-100-12B | 38.2 | 18.6 | 34.8 | 17.0 | 26.1 | 12.2 | 32.2 | 14.5 | 18.3 | 8.7 | 23.9 | 12.9 | 12.5 | 7.0 | 19.8 |
| NLLB-200-1.3B | 41.6 | **35.9** | 34.0 | 28.5 | 23.9 | 19.3 | 32.1 | 25.9 | 14.5 | 13.7 | 27.5 | **24.3** | **29.4** | **25.9** | 26.9 |
| Single-FT-1.6B | 40.1 | 33.0 | 34.1 | 27.6 | 23.5 | 18.0 | 31.6 | 24.9 | 18.0 | 15.1 | 26.3 | 22.2 | 26.5 | 22.8 | 26.0 |
| LegoMT2-435-1.6B | **42.9** | 35.6 | **36.8** | **29.5** | **26.0** | **20.6** | **33.9** | **27.0** | **20.5** | **16.8** | **28.1** | 24.2 | 28.6 | 24.9 | **28.2** |

| Model | En→X | | Pt→X | | Hu→X | | Da→X | | Zh→X | | Sw→X | | Pa→X | | AVG. |
|---|---|---|---|---|---|---|---|---|---|---|---|---|---|---|---|
| NLLB-200-54.5B | 40.3 | 30.6 | 34.2 | 26.4 | 29.3 | 23.0 | 33.5 | 25.5 | 25.3 | 20.4 | 29.0 | 22.9 | 29.9 | 24.8 | 28.2 |
| Flores-175M | 21.2 | 4.8 | 20.3 | 4.4 | 16.4 | 3.4 | 20.2 | 4.1 | 12.4 | 2.7 | 12.9 | 3.2 | 3.2 | 1.1 | 9.3 |
| Flores-615M | 29.8 | 7.0 | 26.4 | 5.8 | 22.4 | 4.8 | 26.7 | 5.6 | 17.7 | 4.1 | 19.4 | 4.8 | 5.4 | 1.6 | 13.0 |
| M2M-100-1.2B | 33.8 | 9.6 | 29.2 | 7.7 | 25.4 | 6.5 | 29.2 | 7.4 | 20.8 | 5.5 | 21.5 | 6.6 | 9.7 | 3.1 | 15.4 |
| M2M-100-12B | 36.2 | 14.0 | 31.1 | 11.6 | 26.9 | 9.6 | 31.0 | 10.9 | 21.8 | 8.4 | 23.8 | 9.9 | 13.7 | 6.6 | 18.3 |
| NLLB-200-1.3B | 36.4 | **28.3** | 30.9 | **24.4** | 25.7 | **20.9** | 30.2 | **23.5** | 21.7 | 18.1 | 25.4 | 21.3 | 25.6 | **22.3** | 25.3 |
| Single-FT-1.6B | 35.8 | 24.6 | 30.2 | 21.0 | 24.9 | 18.2 | 30.1 | 21.2 | 22.0 | 17.0 | 25.0 | 19.5 | 25.1 | 18.6 | 23.8 |
| LegoMT2-435-1.6B | **38.6** | 27.5 | **32.5** | 23.3 | **28.3** | 20.7 | **32.9** | 23.2 | **23.6** | **18.2** | **28.2** | **21.9** | **27.5** | 21.4 | **26.3** |

Table 2: Back-translation evaluation results. Back-translation (*src-trg-srcb*) is an unsupervised evaluation method that involves translating source text to target text *src-trg (S-T, such as **En**→**X**)* and then translating target text back to source text *src-srcb (S-S$_b$, such as **En**→**X**→**En**)*. Lower S-T and higher S-S$_b$ are better. Experimental results demonstrate that LegoMT2 outperforms Single-FT on back-translation performance with almost the same *src-trg (S-T)* score.

| Model | S-T↓ En→X→En | S-S$_b$↑ En→X→En | S-T↓ Pt→X→Pt | S-S$_b$↑ Pt→X→Pt | S-T↓ Hu→X→Hu | S-S$_b$↑ Hu→X→Hu | S-T↓ Da→X→Da | S-S$_b$ Da→X→Da | S-T↓ Zh→X→Zh | S-S$_b$↑ Zh→X→Zh | S-T↓ Mt→X→Mt | S-S$_b$↑ Mt→X→Mt | S-T↓ Pa→X→Pa | S-S$_b$↑ Pa→X→Pa | S-T↓ Lo→X→Lo | S-S$_b$↑ Lo→X→Lo |
|---|---|---|---|---|---|---|---|---|---|---|---|---|---|---|---|---|
| Single-FT | **8.3** | 36.6 | **2.8** | 31.3 | **1.7** | 18.1 | **2.6** | 26.7 | **1.3** | 15.8 | **1.4** | 27.9 | **0.2** | 17.4 | **1.1** | 14.6 |
| LegoMT2 | 9.6 | **43.2** | 3.0 | **37.7** | 1.8 | **22.2** | 2.7 | **33.0** | **1.3** | **20.1** | 1.5 | **35.0** | **0.2** | **22.3** | 1.2 | **18.3** |

| Model | S-T↓ Fr→X→Fr | S-S$_b$↑ Fr→X→Fr | Nl→X→Nl | | Bg→X→Bg | | Sk→X→Sk | | Mk→X→Mk | | Is→X→Is | | Ig→X→Ig | | Li→X→Li | |
|---|---|---|---|---|---|---|---|---|---|---|---|---|---|---|---|---|
| Single-FT | **2.7** | 32.1 | **2.7** | 25.0 | **0.7** | 27.5 | **1.7** | 22.9 | **0.7** | 24.3 | **1.7** | 17.7 | **1.4** | 13.4 | 2.5 | 7.1 |
| LegoMT2 | 2.8 | **38.4** | 2.9 | **31.5** | 0.9 | **31.1** | 1.8 | **28.0** | 0.8 | **30.3** | 1.9 | **22.5** | 1.5 | **14.6** | 2.4 | **9.8** |

| Model | S-T↓ Ja→X→Ja | S-S$_b$↑ Ja→X→Ja | Es→X→Es | | Ar→X→Ar | | Lt→X→Lt | | Fo→X→Fo | | De→X→De | | Uk→X→Uk | | Zu→X→Zu | |
|---|---|---|---|---|---|---|---|---|---|---|---|---|---|---|---|---|
| Single-FT | **0.2** | 16.7 | 4.7 | 27.7 | **0.8** | 17.2 | **1.1** | 19.8 | 1.8 | **13.2** | **2.8** | 23.8 | **0.5** | 20.7 | **1.2** | 16.4 |
| LegoMT2 | 0.3 | **21.6** | **4.1** | **33.0** | **0.8** | **21.8** | 1.2 | **24.5** | **1.6** | 12.9 | 2.9 | **30.8** | 0.6 | **26.9** | 1.5 | **20.4** |

2.5 spBLEU on one-to-many translation. For a fair comparison, we only report results by using the shared global encoder and global decoders for all translation directions. With additional language-specific parameters, LegoMT2 alleviates parameter interference and brings better results. Furthermore, unlike traditional synchronous aggregation methods, we adopt asynchronous aggregation to update global parameters to reduce communication costs and delays. The better results also demonstrate that asynchronous training is an effective method for training massive models.

**LegoMT2 supports 435 languages, the supported language number outperforming all existing open-source multilingual machine translation systems.** To build a fair comparison, we conduct a large-scale multilingual training set. The key challenge lies in balancing the trade-off between knowledge transferring and parameter interference. If not well-handled, involving more languages would result in performance degeneration. Due to the lack of high-quality test translations over 400+ languages, we adopt a practical unsupervised metric, Back-spBLEU to compute the BLEU score between source text and back-translated text. As shown in Table 2, we sample several language-centric results and LegoMT2 demonstrates an improvement in back-translation performance without copying source text issues (comparable src-trg scores).

**Human evaluation results show that the performance of LegoMT2 reaches commercial translators' performance.** We manually assessed the performance of Google Translator, Baidu Translator, LegoMT2, and NLLB-200-1.3B models on Chinese-centric translation tasks. The resulting evaluation scores ranged from 0 to 5. A score of 0 meant that the language was not supported or could not be

Figure 2: Analysis on deferred global parameters. The client is able to use delayed global parameters from other clients for inference without experiencing any decrease in performance. This observation substantiates the notion that the employment of deferred global parameters does not exert a significant influence on model training. The # Param is the total number of a system.

Figure 3: Analysis on $\alpha$ and $\beta$. We analyzed the save/load interval and performed two different settings: 1) save interval of $\alpha = 10$min and load interval of $\beta = 20$min; 2) save interval of $\alpha = 20$min and load interval of $\beta = 10$min, while recording the frequency of setting 1 over setting 2. Results indicate the system's performance is negatively affected by low update frequency.

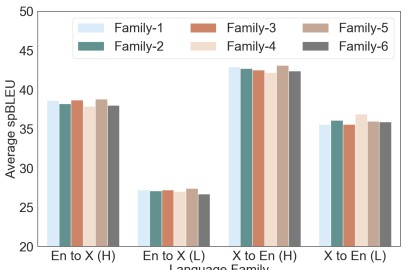

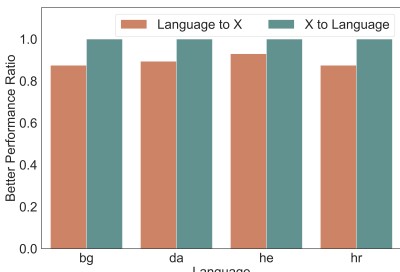

Table 3: The training speed. The number of tokens a model can handle per second is represented by 'Token/s'. The analysis on training demonstrates that LegoMT2 can process more tokens per second with higher GPU efficiency.

| Module | Training Strategy | Parallelism | #Param (during training) | Training Token/s | Speedup |
|---|---|---|---|---|---|
| Single-FT | Centralized Learning | DDP | 1.6B | 76,116 | 40.6× |
| Single-FT + MOE | Centralized Learning | DDP + Pipeline | 12B | 1,873.4 | 1.0× |
| LLaMA | Centralized Learning | DDP + Tensor + Pipeline, Flash Attention | 13B | 7,091.3 | 3.8× |
| LegoMT2 | Traditional Federated Learning | - | 10.4B | 18,719.2 | 10.0× |
| LegoMT2 | Non-blocking Federated Learning | - | 10.4B | 30,280.9 | 16.2× |

translated at all. A score of 5 implied that not only was the content preserved, but the expression was also very smooth. The performance of LegoMT2 is between Google and Baidu, while largely better than NLLB-200-1.3B. More human evaluation details are shown in Appendix D. Among the overlapped languages, LegoMT2 has an average translation score of 3.12, while Google Translator has an average score of 3.64. Among the overlapped languages, LegoMT2's average score is 3.03, while Baidu Translator's average score is 2.55.

**LegoMT2 achieves 1.6× speedups over traditional federated training** Training a single model on multiple GPUs can result in significant communication costs, limiting training efficiency. In this work, we propose LegoMT2 to reduce the bottlenecks caused by aggregation across GPUs. By splitting models into different clients, we can get almost 10× speedups. With reduced communication costs, LegoMT2 further achieves almost 1.6× speedups. Finally, LegoMT2 brings almost 16× speedups. As shown in Table 3, LegoMT2 can process more tokens per second and has higher GPU efficiency than a comparable single model with 12B parameters. We also compared LegoMT2 with widely-used distributed training acceleration frameworks, (e.g., deepspeed (Rajbhandari et al., 2020) and megatron (Shoeybi et al., 2019)), LegoMT2 also shows over 4× throughput improvements. In baseline "Single-FT", we implement DDP and pipeline parallelism (Huang et al., 2019) to accelerate training using the released code training NLLB. In addition, we also report a LLM baseline LLaMA (Touvron et al., 2023) having a similar model size with almost the SOTA distributed setting: DDP + tensor parallelism and pipeline parallelism. Additionally, we use an efficient version of Transformer Flash Attention (Dao et al., 2022) for faster inference. Compared to these advanced training methods, LegoMT2 is a simple but efficient method.

**LegoMT2 brings better performance improvements on high-resource translation** We find that multi-way training benefits high-resource translation by relieving parameter interference. On high-resource translation, LegoMT2 outperforms NLLB-200-1.3B with gains of 1.3 BLEU on many-to-one translation and 2.0 BLEU on one-to-many translation. LegoMT2 largely narrows the gap with the largest machine translation model, NLLB-200-54.5B. Specifically, some results even approach the NLLB-200-54.5B. Taking Family-5 as an example, LegoMT2 yields +3.2% spBLEU improvements over NLLB-200-54.5B on Family-5 on many-to-one settings. Meanwhile, LegoMT2 is on par with

Table 4: Using Dec-Flow, translation performance on *Flores-101* devtest can be improved. Remarkably, this improvement is achievable even for low-resource languages.

| Module | X→Ne | X→Mi | X→Be | X→Km | AVG. |
|---|---|---|---|---|---|
| Mix-Flow | 27.2 | **19.0** | 18.7 | 14.3 | 19.8 |
| Dec-Flow | **28.7** | 18.5 | **20.0** | **16.9** | **21.0** |

Table 5: The experiment results indicate that an extremely unbalanced grouping within the system is not conducive to its optimal performance.

| Direction | Setting | Hr | Bg | Da | AVG. |
|---|---|---|---|---|---|
| LG→X | Similarity | 14.7 | 14.9 | 16.1 | 15.2 |
|  | Random | **16.9** | **17.6** | **18.9** | **17.8** |
| X→LG | Similarity | 12.9 | 18.5 | 19.1 | 16.8 |
|  | Random | **15.0** | **21.2** | **22.4** | **19.5** |

NLLB-200-1.3B on low-resource settings. It is mainly because NLLB focuses on low-resource settings and extremely optimizes low-resource settings based on techniques like back-translation. We only cover limited resources for each translation pair to support more languages.

## 5   ANALYSIS ON LEGOMT2

**Language-specific decoder enhances model performance** According to our results, we find that Dec-Flow largely improves low-resource inference results. To enhance low-resource translation performance, we train Family-7 and Family-8 via Dec-Flows in the second training stage. Table 4 shows that the introduction of Dec-Flow helps low-resource translation.

**Explaining why asynchronous training works** LegoMT2 introduces asynchronous training to reduce communication delays to accelerate training. Each client pulls the latest parameters every $k$ steps and pushes current parameters into the federated server every $m$ steps. It represents that all clients do not always enjoy and latest parameters. To prove whether such delay affects final performance, we conduct experiments by using global modules from other clients for inference.Figure 2 shows delayed global parameters basically do not affect model training. The client can use delayed global parameters from other clients for inference without any performance drops.

**Impact of language groups** In this work, we sort languages based on the size of language-centric data and split languages into different equal-size groups. We adopt this split method because we find that balanced training flows in different clients help multilingual machine translation. In addition, the common strategy of language clustering is by similarity. Therefore, we use similarity clustering to construct a baseline. Given an MNMT model, here we use the single multilingual model to get language id embedding, then directly apply KMeans [4] on those embedding. The clustering results are shown in Appendix E. It is clear that the number of languages in different clusters varies. Meanwhile, we also conduct an experiment by randomly splitting language groups. According to new language groups, we conduct experiments and show results in Table 5. Experiment results show the severely unbalanced distribution of clients hurt the system's performance.

**Impact of save/load ($\alpha/\beta$) intervals setting** Test the effect of different save/load intervals on system performance, i.e. the effect of $\alpha$ and $\beta$ in the algorithm. Theoretically, when $\alpha$ and $\beta$ are small enough, the localized training by LegoMT2 is approximately equal to centralized training. Here, we conduct two different settings: 1) $\alpha = 10$min, $\beta = 20$min and 2) $\alpha = 20$min, $\beta = 40$min. We test two settings LegoMT2 on the *Flores-101* devtest. If the result of setting 1 is better than setting 2, record to 1; otherwise 0. As shown in Figure 3, exchanging information too late may cause the loss of information and reduce the performance of the system.

## 6   CONCLUSION

The typical multilingual neural machine translation is training a single model for all directions with a centralized training schema, which faces many challenges in practice including parameter competition and efficiency problems. In this paper, we propose a new MNMT pre-training framework with federated learning, LegoMT2. Extensive experiments verify the effectiveness of LegoMT2. It brings $16.2\times$ training speedups and large performance gains. We build a translation system that supports 435 languages, the supported language number outperforming all existing open-source multilingual machine translation systems.

---

[4]Implemented by sklearn.

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

LIMITATION

This paper also has several limitations. Firstly, our analysis reveals that the augmentation of low-resource translation through the use of language-specific decoders and encoders is not as effective as anticipated, necessitating a deeper exploration of the interplay between parameter sharing and tension. Secondly, the assessment of few-shot languages continues to pose a significant challenge. Despite our training dataset encompassing 435 languages, our evaluation is limited to back-translation performance, underscoring the need for more rigorous benchmarks.

## A    MECHANISM OF NON-BLOCKING

The non-blocking mechanism is facilitated by asynchronous communication, which effectively minimizes the blocking time caused by communication.

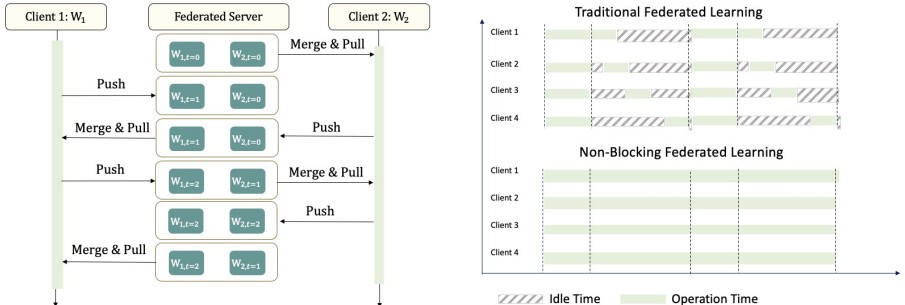

Figure 4: The mechanism of non-blocking. **(1)** Left figure presents an overview of non-blocking communication among different clients. To enable parallel training, LegoMT2 divides the training process into multiple clients, each having its own language-specific data and a copy of global parameters ($W_1$ and $W_2$). To minimize the blocking time caused by communication, we adopt an asynchronous approach. This asynchronous communication ensures that local training is not hindered by waiting for parameter updates. **(2)** The right figure compares traditional federated learning with MT. In traditional federated learning, parameter communication occurs synchronously, which often leads to blocking local training due to the additional synchronous wait.

## B    DATASET CONSTRUCTION

In this section, we will go through the details of constructing a Many-to-Many dataset. The entire pipeline is made up of six steps:

**Step 1: Data Collection**    The unprocessed data is obtained from OPUS[5]. It is an open corpus that collects a large number of parallel sentences from the Web and covers a wide range of domains from legislative to religious texts.

**Step 2: Data Unification**    OPUS has datasets from several sources, which causes the two important problems listed below.

*1) Different Language Code:* Language code is the abbreviation for a language. In OPUS, there are some languages has multiple language codes. One of the causes is that different corpora follow different standards, including ISO 639-1, ISO 639-2, ISO 639-3, or self-defined language codes. Another scenario is that some datasets use language code and region code together. We take ISO 639-1 as the unique code and replaced ISO 639-2 and ISO 639-3 language codes with ISO 639-1 language codes. All these language codes are released by SIL International (formerly known as the Summer Institute of Linguistics)[6].

---

[5]https://opus.nlpl.eu/
[6]https://iso639-3.sil.org/sites/iso639-3/files/downloads/iso-639-3.tab

*2) Inconsistent Operation:* There are some inconsistent operations in some datasets, for example, pre-tokenize for Chinese and Japanese.

To address the above issue, we first handle the case where the language code ends with the region code by removing the region code. Then we standardize all language codes by ISO 639-1. All replaced language codes are listed in Table 6. For the language codes out of ISO 639 series, we report the detail of the language and the corpus that they come from in Table 7. For ease of understanding, we report all used languages with their full name in Table 8. Finally, for the dataset with inconsistent operations, we uniformly perform a removal operation to restore them to natural text.

**Step 3: Data Merging** After unifying the language code and operation, the parallel data with the same language code will be merged into a file.

**Step 4: Data Cleaning** There are some low-quality text in OPUS. They are mainly caused by following reason.

*1) Duplication:* We apply fairseq[7] deduplication script for each language pair.

*2) Missing Translation:* Some low-quality parallel data lacked the correct translation results. We discard using the sentence where the source sentence is without a corresponding target sentence or simply repeat the source sentence as a target sentence.

*3) Length Mismatching:* The length mismatching mainly focuses on the case where the difference between the length of the source and the target is too large. The length of a sentence is defined as the number of words after segmenting with white space (individual characters for Chinese and Japanese). We reuse the filtering script from Moses[8].

**Step 5: Train-Dev-Test Split** The train-dev-test split scheme is specified by the data quantity.

*1) A dataset has over 6.000 parallel sentences.* For a dataset, 2,000 randomly selected parallel sentences are used as a test set, another 2000 randomly selected parallel sentences are used as a validation set, and the rest of the dataset is used as the training set.

*2) A dataset has less than 6.000 parallel sentences.* We use 80%, 10%, and 10% of all parallel sentences as train, validation, and test set.

Meanwhile, we remove the sentence included in the widely used benchmark (WMT, *Flores-101*) from our training and validation set to keep the fairness of comparison.

**Step 6: Data Preprocessing** The data preprocessing consists of two main steps:

*1) Sampling:* Because the full dataset is huge, we sample some data for our training. Our dataset contains 445 languages and about 25B sentence pairs. Table 9 shows the number of parallel sentences in the training set for each language. We present statistics on parallel sentence pairs for the top 100 languages in our constructed data, as shown in Figure 5. The dataset comprises 435 languages and approximately 25 billion sentence pairs. Among these, 19 languages have over 1 billion sentence pairs, while for most languages, the total number of sentence pairs in the dataset does not exceed 1 million.

*2) Preprocessing:* The data is preprocess using the SentencePiece tokenizer provided by Costa-jussà et al. (2022) with a expaned vocabulary of size 491,404.

## C CLIENT INFORMATION

The language group result as shown in Table 10.

**We surprisingly find that low-resource language groups harm pre-training** During the training process of LegoMT2, we include all clients to update global parameters. However, we find that

---

[7] https://github.com/facebookresearch/fairseq/edit/main/examples/backtranslation/deduplicate_lines.py
[8] https://github.com/moses-smt/mosesdecoder

Figure 5: We present an analysis of parallel sentence pairs for the top 100 languages in our constructed dataset. Comprising 435 languages and approximately 25 billion sentence pairs, our dataset reveals that 19 languages have over 1 billion sentence pairs. In contrast, the majority of languages have a total number of sentence pairs that do not exceed 1 million.

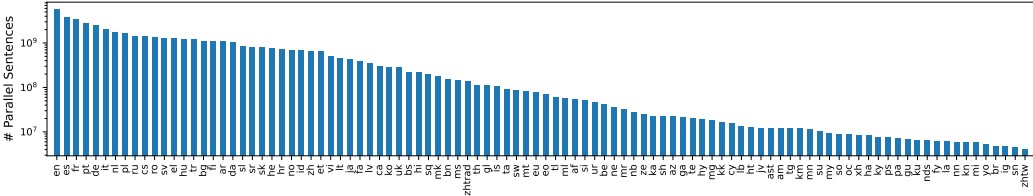

Table 6: Code Replacement List. We use the codes in the column "Original" to replace the codes in the column "replaced" if these replaced codes exist in OPUS.

| Original | Replaced | Original | Replaced | Original | Replaced | Original | Replaced | Original | Replaced | Original | Replaced |
|----------|----------|----------|----------|----------|----------|----------|----------|----------|----------|----------|----------|
| ak | aka | es | es_HN | pt | pt_BR | es | es_CL | kr | kau | tr | tr_TR |
| am | amh | es | es_EC | pt | pt_br | es | es_SV | kv | kpv | ur | ur_PK |
| ar | ara | es | es_CO | pt | pt_PT | es | es_NI | ln | lin | vi | vi_VN |
| ar | ar_SY | fa | fa_IR | rn | run | es | es_UY | mg | mlg | wo | wol |
| ar | ar_TN | fa | fa_AF | rw | kin | es | es_PE | ms | ms_MY | xh | xho |
| ay | aym | ff | ful | sn | sna | es | es_VE | nb | nb_NO | yo | yor |
| az | az_IR | fr | fr_FR | so | som | es | es_AR | nds | nds_nl | ze | ze_zh |
| bg | bg_BG | fr | fr_CA | sr | srp | es | es_MX | nl | nl_NL | ze | ze_en |
| bm | bam | fr | fr_BE | sr | sr_ME | es | es_MX | nl | nl_NL | ze | ze_en |
| bn | bn_IN | fr | fr_ca | st | sot | es | es_PA | nl | nl_BE | zh | zh_cn |
| ca | cat | ha | hau | sw | swa | es | es_CR | nn | nn_NO | zh | zh_CN |
| da | da_DK | hi | hi_IN | ta | ta_LK | es | es_PR | no | no_nb | zhtrad | zh_HK |
| de | de_CH | ig | ibo | tg | tg_TJ | es | es_ES | ny | nya | zhtrad | zh_TW |
| de | de_AT | it | it_IT | ti | tir | es | es_GT | om | orm | zhtrad | zh_tw |
| de | de_DE | jp | jap | tl | tl_PH | es | es_DO | pa | pan | zu | zul |

Table 7: Unkown Language Codes, which are out of ISO 639 series. We can't confirm their full names.

| Code | Dataset | Code | Dataset | Code | Dataset | Code | Dataset | Code | Dataset |
|------|---------|------|---------|------|---------|------|---------|------|---------|
| crp | bible-uedin | cb | MultiCCAligned | sz | MultiCCAligned | sgn | QED | cycl | Tatoeba |
| tc | EUbookshop | cx | MultiCCAligned | zz | MultiCCAligned | iro | QED | nah | Tatoeba |
| zhs | GlobalVoices | ns | MultiCCAligned | ze | OpenSubtitles | mo | QED,Ubuntu | | |
| zht | GlobalVoices | qd | MultiCCAligned | bh | QED | ber | QED,Ubuntu | | |
| tmp | GNOME | qa | MultiCCAligned | bnt | QED | toki | Tatoeba | | |
| gr | GNOME | tz | MultiCCAligned | ry | QED | kzj | Tatoeba | | |

if we directly combine low-resource languages in Family-7 and Family-8 into pre-training, it will increase the proportion of low-resource excessively, thus reducing the performance of the entire system. As shown in Figure 6, we conduct two experiments by involving Family-7 and Faimly-8 or not and report the performance improvements caused by removing Family-7 and Family-8 from pre-training. Experiments show that low-resource languages bring negative effects on pre-training by overestimating the distribution of long-tailed languages.

## D HUMAN EVALUATION PERFORMANCE

Human evaluation results show that the performance of LegoMT2 far exceeds that of Baidu and is on par with Google. We manually assessed the performance of Google Translator, Baidu Translator, LegoMT2, and NLLB-1.3B models on Chinese-centric translation tasks and found that, on average, Google Translator outperformed LegoMT2. LegoMT2 performed better than Baidu Translator and NLLB-1.3B, as shown in Table 11. Here are the specifics of our human evaluation:

1) Data source: We evaluated a total of 100 raw data samples, including 58 samples from the *Flores-101* dataset and 42 samples from the domains of sports, entertainment, and financial news.

Table 8: List of Languages. Our dataset mainly use ISO 639 series as language code. For traditional Chinese, we define "zhtrad" as code.

| Language | Code | Language | Code | Language | Code | Language | Code | Language | Code | Language | Code |
|---|---|---|---|---|---|---|---|---|---|---|---|
| Abkhazian | ab | Corsican | co | Iban | iba | Lower Sorbian | dsb | Ossetian | os | Swahili (macrolanguage) | sw |
| Achinese | ace | Cree | cr | Icelandic | is | Lukpa | dop | Ottoman Turkish (1500-1928) | ota | Swati | ss |
| Achuar-Shiwiar | acu | Creek | mus | Ido | io | Luo (Kenya and Tanzania) | luo | Paite Chin | pck | Swedish | sv |
| Adyghe | ady | Crimean Tatar | crh | Igbo | ig | Lushootseed | lut | Palauan | pau | Swiss German | gsw |
| Afar | aa | Croatian | hr | Iloko | ilo | Luxembourgish | lb | Pali | pi | Syriac | syr |
| Afrihili | afh | Cusco Quechua | quz | Indonesian | id | Luyia | luy | Pampanga | pam | Tachawit | shy |
| Afrikaans | af | Czech | cs | Ingrian | izh | Macedonian | mk | Pangasinan | pag | Tachelhit | shi |
| Aguaruna | agr | Danish | da | Ingush | inh | Macedo-Romanian | rup | Panjabi | pa | Tagal Murut | mvv |
| Ainu (Japan) | ain | Dari | prs | Interlingua | ia | Madurese | mad | Papiamento | pap | Tagalog | tl |
| Akan | ak | Dinka | din | Interlingue | ie | Maithili | mai | Papuan Malay | pmy | Tahaggart Tamahaq | thv |
| Akawaio | ake | Drents | drt | Inuktitut | iu | Malagasy | mg | Pedi | nso | Tahitian | ty |
| Aklanon | akl | Dungan | dng | Inupiaq | ik | Malay (individual language) | zlm | Pennsylvania German | pdc | Tajik | tg |
| Albanian | sq | Dutch | nl | Iranian Persian | pes | Malay (macrolanguage) | ms | Persian | fa | Talossan | tzl |
| Algerian Arabic | arq | Dutton World Speedwords | dws | Irish | ga | Malayalam | ml | Phoenician | phn | Talysh | tly |
| American Sign Language | ase | Dzongkha | dz | Italian | it | Maltese | mt | Picard | pcd | Tamashek | tmh |
| Amharic | am | Eastern Canadian Inuktitut | ike | Jakun | jak | Mam | mam | Piemontese | pms | Tamil | ta |
| Ancient Greek (to 1453) | grc | Eastern Mari | mhr | Jamaican Creole English | jam | Mambae | mgm | Pipil | ppl | Tarifit | rif |
| Ancient Hebrew | hbo | Eastern Maroon Creole | djk | Japanese | ja | Mandarin Chinese | cmn | Plateau Malagasy | plt | Tase Naga | nst |
| Arabic | ar | Efik | efi | Javanese | jv | Manx | gv | Polish | pl | Tatar | tt |
| Aragonese | an | Egyptian Arabic | arz | Jewish Babylonian Aramaic | tmr | Maori | mi | Portuguese | pt | Telugu | te |
| Armenian | hy | Emilian | egl | Kabyle | kab | Marathi | mr | Potawatomi | pot | Tena Lowland Quichua | quw |
| Arpitan | frp | English | en | Kadazan Dusun | dtp | Marshallese | mh | Prussian | prg | Tetelcingo Nahuatl | nhg |
| Asháninka | cni | Erzya | myv | Kalaallisut | kl | Mesopotamian Arabic | acm | Pushto | ps | Tetum | tet |
| Assamese | as | Esperanto | eo | Kalmyk | xal | Miahuatlán Zapotec | zam | Quechua | qu | Thai | th |
| Asturian | ast | Estonian | et | Kamba (Kenya) | kam | Middle English (1100-1500) | enm | Quenya | qya | Tibetan | bo |
| Avaric | av | Evenki | evn | Kannada | kn | Middle French (ca. 1400-1600) | frm | Quiotepec Chinantec | chq | Tigrinya | ti |
| Avestan | ae | Ewe | ee | Kanuri | kr | Mikasuki | mik | Rapanui | rap | Tohono O'odham | ood |
| Awadhi | awa | Extremaduran | ext | Kaqchikel | cak | Mi'kmaq | mic | Romanian | ro | Tok Pisin | tpi |
| Aymara | ay | Faroese | fo | Karelian | krl | Min Dong Chinese | cdo | Romansh | rm | Tonga (Tonga Islands) | to |
| Azerbaijani | az | Fiji Hindi | hif | Kashmiri | ks | Min Nan Chinese | nan | Romany | rom | Traditional Chinese | zhtrad |
| Baluchi | bal | Fijian | fj | Kashubian | csb | Minangkabau | min | Rundi | rn | Tsonga | ts |
| Bambara | bm | Filipino | fil | Kazakh | kk | Mingrelian | xmf | Russian | ru | Tswana | tn |
| Banjar | bjn | Finnish | fi | Kekchí | kek | Mirandese | mwl | Rusyn | rue | Tupí | tpw |
| Barasana-Eduria | bsn | French | fr | Khakas | kjh | Miskito | miq | Samoan | sm | Turkish | tr |
| Bashkir | ba | Friulian | fur | Khasi | kha | Modern Greek (1453-) | el | Samogitian | sgs | Turkmen | tk |
| Basque | eu | Fulah | ff | Khmer | km | Mohawk | moh | Sango | sg | Tuvalu | tvl |
| Bavarian | bar | Galela | gbi | K'iche' | quc | Mongolian | mn | Sanskrit | sa | Twi | tw |
| Baybayanon | bvy | Galician | gl | Kikuyu | kik | Morisyen | mfe | Santali | sat | Uab Meto | aoz |
| Belarusian | be | Gan Chinese | gan | Kinyarwanda | rw | Moroccan Arabic | ary | Sardinian | sc | Udmurt | udm |
| Bemba (Zambia) | bem | Ganda | lg | Kirghiz | ky | Mossi | mos | Saterfriesisch | stq | Uighur | ug |
| Bengali | bn | Garhwali | gbm | Klingon | tlh | Nauru | na | Scots | sco | Ukrainian | uk |
| Berom | bom | Georgian | ka | Koasati | cku | Navajo | nv | Scottish Gaelic | gd | Uma | ppk |
| Bhojpuri | bho | German | de | Kölsch | ksh | Neapolitan | nap | Sediq | trv | Umbundu | umb |
| Bislama | bi | Gheg Albanian | aln | Komi | kv | Nepali (individual language) | npi | Serbian | sr | Upper Sorbian | hsb |
| Bodo (India) | brx | Gilbertese | gil | Komi-Permyak | koi | Nepali (macrolanguage) | ne | Serbo-Croatian | sh | Urdu | ur |
| Bosnian | bs | Goan Konkani | gom | Kongo | kg | Nigerian Fulfulde | fuv | Shan | shn | Uspanteco | usp |
| Breton | br | Gothic | got | Korean | ko | Niuean | niu | Shona | sn | Uzbek | uz |
| Brithenig | bzt | Gronings | gos | Kotava | avk | Nogai | nog | Shuar | jiv | Venda | ve |
| Buginese | bug | Guadeloupean Creole French | gcf | Kriang | ngt | North Levantine Arabic | apc | Shuswap | shs | Venetian | vec |
| Bulgarian | bg | Guarani | gn | Kuanyama | kj | North Moluccan Malay | max | Sicilian | scn | Vietnamese | vi |
| Buriat | bua | Guerrero Amuzgo | amu | Kurdish | ku | Northern Frisian | frr | Silesian | szl | Vlaams | vls |
| Burmese | my | Guerrero Nahuatl | ngu | Kven Finnish | fkv | Northern Kurdish | kmr | Sindarin | sjn | Volapük | vo |
| Cabécar | cjp | Gujarati | gu | Láadan | ldn | Northern Sami | se | Sindhi | sd | Walloon | wa |
| Camsá | kbh | Gulf Arabic | afb | Ladin | lld | Northwestern Ojibwa | ojb | Sinhala | si | Walser | wae |
| Catalan | ca | Haida | hai | Ladino | lad | Norwegian | no | Slovak | sk | Waray (Philippines) | war |
| Cebuano | ceb | Haitian | hat | Lakota | lkt | Norwegian Bokmål | nb | Slovenian | sl | Welsh | cy |
| Central Huasteca Nahuatl | nch | Hakha Chin | cnh | Lao | lo | Norwegian Nynorsk | nn | Somali | so | Western Frisian | fy |
| Central Kurdish | ckb | Hakka Chinese | hak | Latgalian | ltg | Novial | nov | South Azerbaijani | azb | Western Panjabi | pnb |
| Central Sama | sml | Hausa | ha | Latin | la | Nuer | nus | South Ndebele | nr | Wolaytta | wal |
| Chamorro | ch | Hawaiian | haw | Latvian | lv | Nyanja | ny | Southern Kurdish | sdh | Wolof | wo |
| Chavacano | cbk | Hebrew | he | Ligurian | lij | Occitan (post 1500) | oc | Southern Sami | sma | Wu Chinese | wuu |
| Chechen | ce | Hiligaynon | hil | Limburgan | li | Old English (ca. 450-1100) | ang | Southern Sotho | st | Xhosa | xh |
| Cherokee | chr | Hindi | hi | Lingala | ln | Old French (842-ca. 1400) | fro | Southwestern Dinka | dik | Yakut | sah |
| Chhattisgarhi | hne | Hiri Motu | ho | Lingua Franca Nova | lfn | Old Frisian | ofs | Spanish | es | Yaqui | yaq |
| Chinese | zh | Hmong Daw | mww | Literary Chinese | lzh | Old Norse | non | Standard Malay | zsm | Yiddish | yi |
| Choctaw | cho | Ho | hoc | Lithuanian | lt | Old Russian | orv | Standard Moroccan Tamazight | zgh | Yoruba | yo |
| Church Slavic | cu | Huastec | hus | Liv | liv | Old Spanish | osp | Sumerian | sux | Zarma | dje |
| Chuvash | cv | Hungarian | hu | Lojban | jbo | Oriya (macrolanguage) | or | Sundanese | su | Zaza | zza |
| Coptic | cop | Hunsrik | hrx | Lombard | lmo | Orizaba Nahuatl | nlv | Swabian | swg | Zulu | zu |
| Cornish | kw | Hupa | hup | Low German | nds | Oromo | om | Swahili (individual language) | swh | | |

2) Annotation method: To better evaluate the quality of large-scale translation, we adopted a translation and back-translation method in our human evaluation. For instance, we presented a Chinese input text to the models and asked them to produce a translated text and a back-translated Chinese text. The annotators assessed the degree of information overlap between the input text and the back-translated Chinese text.

3) Annotation process: To ensure inter-annotator agreement, we assigned each sample to two distinct annotators at a cost of $0.028 per datum. The resulting evaluation scores ranged from 0 to 5. A score of 0 meant that the language was not supported or could not be translated at all. A score of 5 implied that not only was the content preserved, but the expression was also very smooth. The average inter-annotator agreement score was 0.79, indicating good evaluation quality.

Among the overlapping languages, LegoMT2 had an average translation score of 3.12, while Google Translator had an average score of 3.64. Among the non-overlapping languages, LegoMT2's average score was 3.03, while Baidu Translator's average score was 2.55.

## E    LANGUAGE GROUP BY KMEANS.

In this study, we categorize languages based on the magnitude of language-specific data and partition them into distinct groups of equivalent size. This partitioning method was chosen due to our observation that balanced training flows among different clients facilitate multilingual machine translation. Furthermore, language clustering is commonly performed based on similarity. While it is possible to utilize existing linguistic knowledge for classification, this approach becomes labor-intensive when dealing with more than 400 languages. As such, we employ similarity clustering

Table 9: Statistics of the constructed dataset.

| code | sentence pairs | code | sentence pairs | code | sentence pairs | code | sentence pairs | code | sentence pairs | code | sentence pairs |
|---|---|---|---|---|---|---|---|---|---|---|---|
| aa | 25190 | cni | 366213 | he | 768039586 | lo | 2934940 | pag | 41 | swg | 1485 |
| ab | 24734 | co | 5679 | hi | 218864052 | lt | 467441039 | pam | 1897 | swh | 767 |
| ace | 55744 | cop | 392273 | hif | 30 | ltg | 25791 | pap | 16428 | syr | 393273 |
| acm | 38 | cr | 128 | hil | 2044 | luo | 91 | pau | 28 | sz | 10 |
| acu | 275510 | crh | 583965 | hne | 3624732 | lut | 61 | pck | 1722862 | szl | 45989 |
| ady | 10 | crp | 1698290 | ho | 51 | luy | 105 | pdc | 63 | ta | 90971643 |
| ae | 139 | cs | 1457869889 | hoc | 517 | lv | 355693685 | pes | 1744278 | tc | 2831 |
| af | 55335682 | csb | 1087185 | hr | 737162068 | lzh | 540 | phn | 30 | te | 20088988 |
| afb | 77 | cu | 1996 | hrx | 558 | mad | 947 | pi | 2306 | tet | 12255 |
| afh | 73 | cv | 24927 | hsb | 662844 | mai | 1969608 | pl | 1650606708 | tg | 11994239 |
| agr | 296459 | cx | 2852903 | ht | 12715844 | mam | 358606 | plt | 1715974 | th | 111068105 |
| ain | 306 | cy | 15839521 | hu | 1254849755 | max | 345 | pms | 6128 | thv | 41 |
| ak | 13593 | cycl | 43 | hup | 287 | mfe | 8944 | pmy | 5324 | ti | 98816 |
| ake | 278088 | da | 1024948205 | hus | 81 | mg | 18564176 | pnb | 154 | tk | 237791 |
| akl | 23 | de | 2564377381 | hy | 19095048 | mgm | 27 | pot | 163018 | tl | 62019683 |
| aln | 23 | dik | 290563 | ia | 243295 | mh | 188 | ppk | 363985 | tlh | 22430 |
| am | 12065296 | din | 2457 | iba | 42 | mhr | 150906 | ppl | 27 | tly | 38 |
| amu | 375783 | dje | 1728497 | id | 697068570 | mi | 5753968 | prg | 407 | tmh | 166643 |
| an | 457768 | djk | 354595 | ie | 19196 | mic | 10 | prs | 14123 | tmp | 19110 |
| ang | 151166 | dng | 22 | ig | 4802381 | mik | 15 | ps | 7700300 | tmr | 380 |
| aoz | 20 | dop | 381489 | ik | 393 | min | 84 | pt | 2812386990 | tn | 488012 |
| apc | 35 | drt | 46 | ike | 32 | miq | 8506 | qa | 521 | to | 1479 |
| ar | 1079338710 | dsb | 7157 | ilo | 891090 | mk | 177474445 | qd | 1896 | toki | 37627 |
| arq | 50647 | dtp | 1911 | inh | 17366 | ml | 57004885 | qu | 31780 | tpi | 81 |
| ary | 155 | dws | 56 | io | 149762 | mn | 11603195 | quc | 358962 | tpw | 72 |
| arz | 78593 | dz | 161086 | iro | 8 | mo | 31 | quw | 391236 | tr | 1193231266 |
| as | 2307772 | ee | 376963 | is | 104661362 | moh | 72 | quz | 20 | trv | 1535 |
| ase | 6084 | efi | 4358 | it | 2093054002 | mos | 1864 | qya | 171 | ts | 51109 |
| ast | 12083731 | egl | 322 | iu | 6120 | mr | 31855664 | rap | 22 | tt | 1501339 |
| av | 7398 | el | 1258104866 | izh | 9 | ms | 149607728 | rif | 60 | tvl | 13 |
| avk | 1757 | enm | 741 | ja | 434118540 | mt | 82700941 | rm | 10037 | tw | 479 |
| awa | 225 | eo | 71211656 | jak | 368614 | mus | 9229 | rn | 6358 | ty | 17 |
| ay | 43034 | es | 3911731697 | jam | 29 | mvv | 8 | ro | 1335221001 | tz | 55 |
| az | 22317802 | et | 647382971 | jbo | 53616 | mwl | 36153 | rom | 391669 | tzl | 1415 |
| azb | 6270 | eu | 79865761 | jiv | 278960 | mww | 65 | ru | 1460007489 | udm | 53 |
| ba | 414706 | evn | 64 | jv | 12235804 | my | 9517618 | rue | 175 | ug | 915049 |
| bal | 2285 | ext | 57 | ka | 23136675 | myv | 22 | rup | 2965 | uk | 280561930 |
| bar | 75324 | fa | 383151473 | kab | 469669 | na | 16 | rw | 1271784 | umb | 54 |
| be | 41361204 | ff | 329791 | kam | 8 | nah | 160 | ry | 5054 | ur | 47703807 |
| bem | 19058 | fi | 1081684445 | kbh | 407244 | nan | 9666 | sa | 93931 | usp | 368078 |
| ber | 192407 | fil | 1091348 | kek | 1674772 | nap | 3093 | sah | 835 | uz | 3381954 |
| bg | 1130459221 | fj | 3443 | kg | 131420 | nb | 27802066 | sat | 114 | ve | 8057 |
| bh | 2613 | fkv | 498 | kha | 1282 | nch | 75 | sc | 55166 | vec | 26482 |
| bho | 1263 | fo | 228021 | kik | 267 | nds | 6525803 | scn | 7790 | vi | 500458007 |
| bi | 6112 | fr | 3412558369 | kj | 5446 | ne | 36233624 | sco | 44793 | vls | 430 |
| bjn | 16 | frm | 827 | kjh | 15 | ngt | 15 | sd | 2816050 | vo | 4484 |
| bm | 5993 | fro | 44 | kk | 16875999 | ngu | 31 | sdh | 28 | wa | 2659876 |
| bn | 156924699 | frp | 82087 | kl | 33411 | nhg | 376653 | se | 1912829 | wae | 74267 |
| bnt | 1534 | frr | 402 | km | 11875237 | niu | 24 | sg | 10 | wal | 374085 |
| bo | 108249 | fur | 328314 | kmr | 714 | nl | 1777745084 | sgn | 688 | war | 1230 |
| bom | 39 | fuv | 2482 | kn | 5999187 | nlv | 12 | sgs | 40 | wo | 983607 |
| br | 4839927 | fy | 6208767 | ko | 285583000 | nn | 6036066 | sh | 22711333 | wuu | 10993 |
| brx | 2126 | ga | 21763185 | koi | 12 | no | 698491446 | shi | 378312 | xal | 3583 |
| bs | 221212239 | gan | 12 | kr | 11412 | nog | 79 | shn | 40453 | xh | 8640822 |
| bsn | 325256 | gbi | 350547 | krl | 314 | non | 16 | shs | 20833 | xmf | 36 |
| bua | 1948 | gbm | 33 | ks | 64356 | nov | 919 | shy | 15 | yaq | 81 |
| bug | 1659 | gcf | 1009 | ksh | 2892 | npi | 93 | si | 52111630 | yi | 1038001 |
| bvy | 21 | gd | 833984 | ku | 6566496 | nr | 874 | sjn | 293 | yo | 5433688 |
| bzt | 1196 | gil | 12 | kv | 59 | ns | 103879 | sk | 809520471 | zam | 1379 |
| ca | 303844363 | gl | 110969736 | kw | 82917 | nso | 427594 | sl | 834996012 | ze | 25667080 |
| cak | 355513 | gn | 10158 | ky | 7814500 | nst | 644 | sm | 73 | zgh | 97 |
| cb | 354133 | gom | 49256 | kzj | 1543 | nus | 2496 | sma | 39 | zh | 660697725 |
| cbk | 2141 | gos | 3382 | la | 6202902 | nv | 358 | sml | 1711 | zhs | 37264 |
| cdo | 20 | got | 234 | lad | 4634 | ny | 4130938 | sn | 4557031 | zht | 39547 |
| ce | 13338 | gr | 5607 | lb | 13159469 | oc | 8708362 | so | 9082662 | zhtrad | 143676341 |
| ceb | 3534028 | grc | 1105 | ldn | 163 | ofs | 8 | sq | 203389893 | zlm | 92 |
| ch | 356694 | gsw | 247 | lfn | 13823 | ojb | 299926 | sr | 825444520 | zsm | 2719 |
| cho | 309 | gu | 7015993 | lg | 248315 | om | 203313 | ss | 672164 | zu | 3516139 |
| chq | 356343 | gv | 537765 | li | 365187 | ood | 21 | st | 10364 | zz | 44 |
| chr | 392260 | ha | 8504550 | lij | 1673 | or | 1005953 | stq | 128 | zza | 27246 |
| cjp | 389090 | hai | 1866 | liv | 27 | orv | 1348 | su | 10421463 | | |
| ckb | 78358 | hak | 16 | lkt | 25 | os | 61302 | sux | 153 | | |
| cku | 571 | haw | 385 | lld | 10268 | osp | 10 | sv | 1297167012 | | |
| cmn | 16159 | hbo | 101 | lmo | 13318 | ota | 880 | sw | 87842873 | | |
| cnh | 1784 | | | ln | 171241 | pa | 7181860 | | | | |

to establish a baseline. Utilizing a single multilingual model, we obtain language id embeddings and apply KMeans clustering to them. The results of this clustering are depicted in Figure 7, which clearly illustrates the variation in the number of languages across different clusters. We also conduct an experiment in which language groups are randomly split. Our findings, indicate that a severely unbalanced distribution of clients negatively impacts system performance.

A comparison is made between the performance of ChatGPT and LegoMT2 using the first 100 samples extracted from the *Flores-101* devtest. The effects of both X→En and En→X are tested. For the sys-

Table 10: Language groups. We sort languages based on the size of language-centric data and split them into 8 equal-size chunks.

| Family | 435 Languages |
|---|---|
| Family-1 | fr, es, en |
| Family-2 | nl, tr, pl, it, de, pt |
| Family-3 | bg, ar, ru, fa, el, hu, ro, cs |
| Family-4 | sk, da, uk, sl, he, fi, id, sv, vi |
| Family-5 | ko, sq, hr, mk, sr, zh, no, bs, hi |
| Family-6 | eo, mt, eu, sw, is, lv, ca, th, ms, zhtrad, bn, lt, et |
| Family-7 | ig, km, ky, ps, tg, gv, nb, br, ss, sh, ze, zu, nn, pa, so, sn, kk, cy, mg, am, xh, az, gu , hy, kn, te, ga, gl, be, mr, ne, si, af, ml, tl |
| Family-8 | iro, kam, mvv, ofs, izh, ady, mic, osp, sg, sz, gan, gil, koi, nlv, tvl, kjh, mik, ngt, shy, bjn, hak, na, non, ty, aoz, cdo, quz, bvy, ood, dng, myv, rap, akl, aln, niu, lkt, liv, mgm, ppl, pau, sdh, jam, hif, phn, mo, ngu, ike, gbm, apc, xmf, acm, tly, bom, sma, sgs, pag, thv, iba, cycl, fro, zz, drt, ho, udm, umb, tz, dws, ext, kv, rif, lut, pdc, evn, mww, moh, tpw, afh, sm, nch, afb, nog, hus, tpi, yaq, min, luo, zlm, npi, zgh, hbo, luy, sat, cr, stq, ae, sux, pnb, ary, nah, ldn, qya, rue, mh, awa, got, pcd, gsw, kik, hup, sjn, ain, cho, krl, egl, max, nv, tmr, haw, ik, frr, prg, vls, tw, fkv, hoc, qa, lzh, hrx, cku, nst, sgn, kmr, enm, swh, frm, sah, nr, ota, nov, mad, gcf, grc, bzt, war, bho, kha, orv, que, zam, tzl, to, swg, bnt, trv, kzj, bug, lij, sml, avk, cnh, mos, hai, qd, pam, dtp, bua, cu, hil, brx, cbk, zhyue, bal, pi, din, fuv, nus, bh, zsm, tc, ksh, rup, nap, gos, fj, xal, efi, vo, lad, ry, pmy, kj, gr, co, bm, ase, bi, iu, pms, azb, rn, hbs, dsb, av, scn, ve, miq, mfe, mus, mwl, nan, rm, gn, lld, st, wuu, kr, tet, lmo, ce, ak, lfn, prs, cmn, pap, ber, inh, bem, tmp, ie, toki, shs, tlh, ab, cv, aa, ltg, zhs, vec, zza, zht, qu, kl, ilo, bar, shn, ay, sco, szl, arz, gom, arq, ts, jbo, sc, ace, os, ks, wae, ckb, frp, kw, zhtw, ti, sa, ns, bo, kg, ba, fo, io, dz, mhr, ang, ln, pot, tmh, om, fil, ia, lg, tk, csb, yi, acu, ake, cb, jiv, se, dik, an, tn, agr, tt, kek, ojb, crp, pck, plt, dje, pes, lb, gbi, djk, cak, mai, bsn, chq, quc, mam, ch, fur, ppk, cni, usp, jak, wal, amu, ee, lo, rw, nhg, shi, dop, wa, cx, li, cjp, rom, quw, chr, cop, syr, ug, su, kab, hsb, kbh, hne, uz, nso, fy, ht, wo, crh, la, ny, or, gd, oc, jv, nds, mn, as, ast |

Figure 6: Pre-training is negatively impacted by low-resource language groups. Two experiments were conducted to determine the effects of including or excluding Family-7 and Family-8. The Y-axis displays the performance improvements from pre-training.

Figure 7: Language clustering results. After obtaining the model through single-model fine-tuning, we extract embedding vectors corresponding to all language IDs. Then perform K-means clustering on this embedding matrix and visualize the clustering results using PCA. The results show that the clustering quantity is unbalanced between clusters.

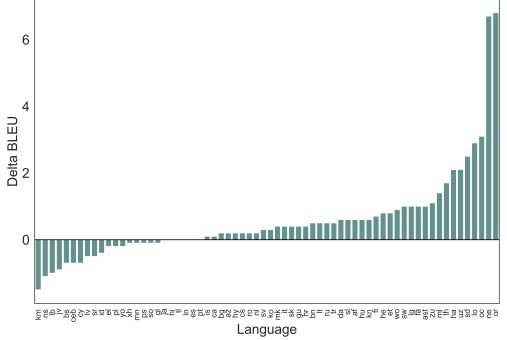

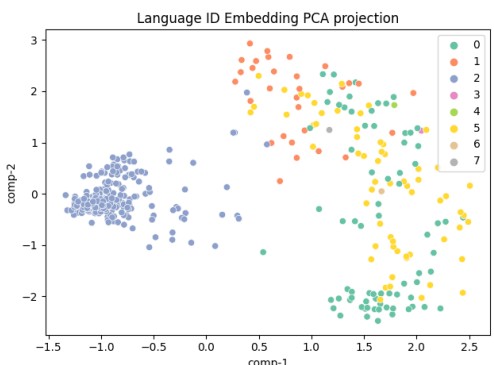

tem, the given prompt for ChatGPT is: "You are a helpful assistant that translates {SOURCE_LANG} to {TARGET_LANG}." For the sentences that needed to be translated, the given prompt is: "Translate the following {SOURCE_LANG} text to {TARGET_LANG}: {SOURCE_TEXT}." Both zero-shot and eight-shot results are tested, with the eight-shot samples being randomly extracted from the *Flores-101* dev.

Table 11: Human evaluation result on Zh→X direction. Manually comparing the performance of LegoMT2 with NLLB-200-1.3B, Google, and Baidu translators, respectively. Based on the results of human evaluation, it has been found that LegoMT2's performance surpasses that of Baidu by a substantial margin and is on par with Google's performance.

| Translator | En | Es | Fr | Pt | De | It | Nl | Pl | Ru | Da | Kn | Mr | Ka | Ja | Fa |
|---|---|---|---|---|---|---|---|---|---|---|---|---|---|---|---|
| LegoMT2 | 3.96 | 2.99 | 3.34 | 3.74 | 3.61 | 3.53 | 2.94 | 3.58 | 3.64 | 3.63 | 3.29 | 3.57 | 2.78 | 3.63 | 2.90 |
| NLLB-200-1.3B | 2.52 | 2.16 | 2.31 | 2.44 | 2.43 | 2.10 | 1.86 | 2.10 | 2.39 | 2.16 | 2.52 | 2.57 | 1.87 | 2.30 | 2.32 |
| Google | 4.32 | 3.33 | 3.46 | 3.95 | 3.88 | 3.96 | 3.32 | 3.82 | 3.84 | 4.25 | 3.80 | 3.96 | 3.81 | 3.66 | 3.10 |
| Baidu | 4.32 | 3.18 | 3.51 | 3.94 | 3.87 | 3.93 | 3.30 | 3.72 | 3.80 | 4.02 | 1.95 | 2.70 | 1.95 | 4.14 | 2.79 |
| Correlation | 0.70 | 0.44 | 0.41 | 0.68 | 0.62 | 0.79 | 0.72 | 0.69 | 0.7 | 0.71 | 0.83 | 0.51 | 0.83 | 0.76 | 0.27 |

| Translator | Sr | Sk | He | Hr | No | Id | Et | Vi | Lt | Ms | Yo | Te | Hy | Ca | Ko |
|---|---|---|---|---|---|---|---|---|---|---|---|---|---|---|---|
| LegoMT2 | 3.23 | 3.28 | 3.54 | 3.47 | 2.70 | 3.76 | 2.87 | 3.95 | 3.69 | 3.41 | 3.18 | 3.46 | 3.48 | 3.45 | 3.24 |
| NLLB-200-1.3B | 2.27 | 1.88 | 1.88 | 1.86 | 2.03 | 2.49 | 2.02 | 2.70 | 2.33 | 2.34 | 2.44 | 2.32 | 2.12 | 2.47 | 2.35 |
| Google | 3.74 | 3.72 | 3.90 | 3.70 | 3.32 | 3.87 | 3.16 | 4.23 | 4.06 | 4.00 | 3.74 | 4.01 | 3.59 | 3.72 | 3.69 |
| Baidu | 3.26 | 3.29 | 3.39 | 3.55 | 2.62 | 3.73 | 3.15 | 4.10 | 3.77 | 3.63 | 1.67 | 1.87 | 3.29 | 3.48 | 4.20 |
| Correlation | 0.62 | 0.60 | 0.72 | 0.7 | 0.66 | 0.64 | 0.45 | 0.63 | 0.65 | 0.6 | 0.74 | 0.76 | 0.71 | 0.56 | 0.76 |

| Translator | Th | Gl | Is | Mt | Tl | Ml | Af | Ur | Be | Tg | Ig | Kk | Cy | Uk | Bs |
|---|---|---|---|---|---|---|---|---|---|---|---|---|---|---|---|
| LegoMT2 | 3.49 | 2.80 | 3.28 | 3.44 | 3.40 | 3.58 | 3.39 | 3.13 | 1.60 | 3.51 | 3.29 | 3.30 | 3.14 | 3.55 | 3.32 |
| NLLB-200-1.3B | 2.18 | 2.08 | 1.92 | 2.54 | 2.36 | 2.72 | 1.71 | 2.72 | 1.91 | 2.37 | 2.20 | 2.16 | 2.20 | 2.31 | 2.08 |
| Google | 3.75 | 2.94 | 4.09 | 4.14 | 3.94 | 4.05 | 3.63 | 3.99 | 3.26 | 3.94 | 3.42 | 3.73 | 3.43 | 3.91 | 3.79 |
| Baidu | 3.91 | 2.93 | 3.12 | 3.82 | 3.25 | 2.77 | 3.03 | 2.81 | 2.89 | 2.54 | 2.23 | 0.00 | 3.05 | 3.45 | 3.59 |
| Correlation | 0.66 | 0.47 | 0.77 | 0.61 | 0.65 | 0.68 | 0.6 | 0.63 | 0.69 | 0.65 | 0.69 | 0.94 | 0.61 | 0.66 | 0.74 |

| Translator | Km | My | So | Oc | Xh | Ha | Ky | Pa | Gu | Ln | Sn | Jv | Ast | Hi | Mk |
|---|---|---|---|---|---|---|---|---|---|---|---|---|---|---|---|
| LegoMT2 | 2.92 | 2.83 | 2.75 | 2.82 | 3.52 | 2.98 | 3.19 | 3.37 | 3.49 | 3.21 | 2.85 | 3.26 | 2.41 | 3.37 | 3.71 |
| NLLB-200-1.3B | 2.15 | 1.86 | 2.10 | 2.55 | 2.61 | 2.40 | 1.98 | 2.73 | 2.60 | 2.45 | 2.20 | 2.40 | 1.36 | 2.55 | 2.59 |
| Google | 3.43 | 3.20 | 3.12 | 0.00 | 3.83 | 3.44 | 3.83 | 3.92 | 3.74 | 3.88 | 3.22 | 3.70 | 0.00 | 3.73 | 4.15 |
| Baidu | 1.75 | 1.90 | 1.99 | 2.67 | 2.85 | 1.89 | 2.31 | 1.91 | 2.13 | 1.09 | 1.83 | 1.12 | 3.29 | 2.48 | 3.84 |
| Correlation | 0.83 | 0.74 | 0.45 | 0.92 | 0.58 | 0.6 | 0.78 | 0.72 | 0.71 | 0.82 | 0.53 | 0.88 | 0.85 | 0.63 | 0.67 |

| Translator | Cs | Ro | Sv | El | Hu | Tr | Bg | Fi | Ar | Lg | Ny | Am | Lo | Bn | As |
|---|---|---|---|---|---|---|---|---|---|---|---|---|---|---|---|
| LegoMT2 | 3.50 | 3.41 | 3.42 | 3.64 | 3.72 | 3.84 | 3.06 | 2.97 | 3.18 | 2.24 | 2.86 | 3.07 | 3.60 | 3.42 | 2.64 |
| NLLB-200-1.3B | 2.09 | 2.40 | 2.15 | 2.24 | 2.03 | 2.21 | 1.55 | 2.16 | 1.62 | 1.98 | 2.34 | 1.99 | 2.84 | 2.42 | 2.00 |
| Google | 3.75 | 3.80 | 3.99 | 3.94 | 3.80 | 4.04 | 3.34 | 3.24 | 3.45 | 3.66 | 3.08 | 3.30 | 3.97 | 3.68 | 3.41 |
| Baidu | 3.75 | 1.93 | 3.85 | 3.78 | 3.76 | 3.44 | 3.27 | 3.13 | 2.93 | 2.20 | 1.01 | 1.74 | 1.73 | 2.59 | 1.72 |
| Correlation | 0.72 | 0.72 | 0.64 | 0.61 | 0.69 | 0.59 | 0.59 | 0.28 | 0.54 | 0.83 | 0.83 | 0.49 | 0.59 | 0.68 | 0.63 |

Table 12: Comparison of ChatGPT and LegoMT2: While ChatGPT outperforms LegoMT2 for some language pairs, LegoMT2 has an absolute advantage for the vast majority. On average, ChatGPT lags behind LegoMT2 in both the En→X and X→En directions by more than 6 points.

| X→En | ChatGPT | LegoMT2 | X→En | ChatGPT | LegoMT2 | X→En | ChatGPT | LegoMT2 | X→En | ChatGPT | LegoMT2 |
|---|---|---|---|---|---|---|---|---|---|---|---|
| af | 54.9 | 58.9 | gu | 20.0 | 39.1 | lo | 9.8 | 37.3 | ru | 32.6 | 36.8 |
| am | 2.7 | 32.4 | ha | 13.4 | 31.3 | lt | 30.9 | 35.4 | sd | 13.0 | 22.0 |
| ar | 33.7 | 41.6 | he | 32.6 | 41.5 | luo | 8.1 | 27.5 | sk | 35.5 | 41.6 |
| as | 12.9 | 31.1 | hi | 33.9 | 47.1 | lv | 30.5 | 35.7 | sl | 33.7 | 36.7 |
| ast | 38.3 | 33.3 | hr | 36.9 | 39.5 | mi | 19.4 | 30.0 | sn | 13.2 | 30.5 |
| az | 18.7 | 27.7 | hu | 32.6 | 35.6 | mk | 37.6 | 43.0 | so | 14.3 | 32.5 |
| be | 19.2 | 19.9 | hy | 14.5 | 39.0 | ml | 18.5 | 41.0 | sr | 35.2 | 40.7 |
| bg | 37.1 | 41.4 | id | 40.1 | 45.0 | mn | 11.1 | 30.2 | sv | 46.3 | 49.4 |
| bn | 21.2 | 38.5 | ig | 8.7 | 28.4 | mr | 20.4 | 39.6 | sw | 40.4 | 47.0 |
| bs | 41.3 | 44.6 | is | 28.9 | 35.0 | ms | 43.8 | 47.6 | ta | 13.6 | 32.5 |
| ca | 43.1 | 46.3 | it | 34.4 | 35.5 | mt | 42.8 | 60.5 | te | 18.6 | 42.1 |
| ceb | 37.5 | 45.0 | ja | 26.5 | 30.5 | my | 2.8 | 30.2 | tg | 13.4 | 32.9 |
| cs | 38.2 | 43.7 | jv | 27.4 | 45.1 | ne | 21.2 | 40.5 | th | 21.9 | 33.6 |
| cy | 44.0 | 54.6 | ka | 12.1 | 27.6 | nl | 34.8 | 36.2 | tl | 41.9 | 51.4 |
| da | 47.6 | 51.3 | kam | 9.8 | 19.6 | no | 41.3 | 45.7 | tr | 36.0 | 39.0 |
| de | 41.4 | 44.4 | kea | 33.7 | 51.2 | ns | 13.9 | 43.2 | uk | 37.2 | 41.5 |
| el | 33.9 | 39.1 | kk | 18.6 | 35.6 | ny | 15.1 | 32.4 | umb | 5.0 | 14.8 |
| es | 29.9 | 31.4 | km | 13.6 | 36.8 | oc | 45.3 | 56.8 | ur | 24.6 | 38.5 |
| et | 35.9 | 38.7 | kn | 20.0 | 35.1 | om | 4.9 | 22.6 | uz | 19.3 | 34.6 |
| fa | 30.4 | 37.2 | ko | 26.1 | 28.3 | or | 14.0 | 36.3 | vi | 33.3 | 42.0 |
| ff | 7.3 | 12.0 | ku | 9.6 | 35.7 | pa | 24.0 | 44.2 | wo | 8.5 | 21.5 |
| fi | 31.5 | 33.9 | ky | 10.7 | 26.9 | pl | 29.9 | 33.6 | xh | 17.1 | 39.6 |
| fr | 43.9 | 46.9 | lb | 39.6 | 45.5 | ps | 10.6 | 35.6 | yo | 9.8 | 26.0 |
| ga | 33.2 | 43.3 | lg | 11.1 | 23.1 | pt | 47.5 | 50.5 | zh | 28.3 | 30.5 |
| gl | 39.2 | 40.5 | ln | 10.6 | 28.7 | ro | 42.7 | 48.1 | zu | 18.0 | 41.5 |
| **EN→X** | **ChatGPT** | **LegoMT2** | **EN→X** | **ChatGPT** | **LegoMT2** | **EN→X** | **ChatGPT** | **LegoMT2** | **EN→X** | **ChatGPT** | **LegoMT2** |
| af | 44.3 | 45.3 | gu | 19.0 | 34.8 | lo | 4.0 | 28.9 | ru | 36.0 | 39.0 |
| am | 2.9 | 26.9 | ha | 8.1 | 26.9 | lt | 27.2 | 33.5 | sd | 8.4 | 33.3 |
| ar | 31.6 | 36.1 | he | 27.0 | 37.2 | luo | 4.2 | 18.3 | sk | 34.5 | 38.9 |
| as | 7.3 | 24.6 | hi | 29.2 | 46.6 | lv | 27.9 | 23.2 | sl | 32.5 | 37.1 |
| ast | 29.8 | 30.3 | hr | 34.4 | 35.7 | mi | 16.0 | 19.8 | sn | 5.8 | 19.5 |
| az | 11.8 | 20.4 | hu | 27.2 | 34.9 | mk | 33.1 | 43.4 | so | 6.4 | 18.2 |
| be | 16.4 | 23.4 | hy | 10.5 | 33.1 | ml | 12.0 | 38.0 | sr | 1.5 | 29.3 |
| bg | 38.7 | 49.3 | id | 45.4 | 46.6 | mn | 5.5 | 18.8 | sv | 46.5 | 46.6 |
| bn | 18.4 | 33.7 | ig | 6.2 | 19.9 | mr | 10.4 | 27.4 | sw | 37.5 | 40.1 |
| bs | 34.0 | 35.1 | is | 22.0 | 30.2 | ms | 39.2 | 47.2 | ta | 10.2 | 20.8 |
| ca | 46.8 | 48.9 | it | 35.8 | 36.5 | mt | 31.6 | 64.4 | te | 13.2 | 41.6 |
| ceb | 24.5 | 18.9 | ja | 29.7 | 33.5 | my | 2.5 | 15.5 | tg | 11.0 | 32.5 |
| cs | 36.7 | 40.5 | jv | 15.6 | 30.3 | ne | 15.0 | 26.4 | th | 22.1 | 21.1 |
| cy | 44.0 | 43.8 | ka | 11.1 | 23.0 | nl | 31.7 | 31.9 | tl | 31.2 | 34.9 |
| da | 45.4 | 45.5 | kam | 4.9 | 7.4 | no | 36.6 | 37.2 | tr | 34.5 | 36.4 |
| de | 40.3 | 41.7 | kea | 11.5 | 17.5 | ns | 6.6 | 26.8 | uk | 33.3 | 40.1 |
| el | 30.9 | 34.5 | kk | 11.1 | 33.7 | ny | 6.3 | 23.8 | umb | 2.8 | 2.9 |
| es | 32.3 | 30.7 | km | 4.4 | 15.9 | oc | 28.2 | 44.0 | ur | 16.8 | 27.4 |
| et | 33.6 | 34.4 | kn | 14.3 | 31.6 | om | 1.7 | 10.8 | uz | 15.8 | 27.3 |
| fa | 25.4 | 35.0 | ko | 25.0 | 26.0 | or | 11.3 | 32.3 | vi | 38.7 | 43.2 |
| ff | 3.0 | 0.1 | ku | 5.0 | 3.5 | pa | 20.3 | 36.1 | wo | 5.1 | 6.3 |
| fi | 33.3 | 31.2 | ky | 7.2 | 24.4 | pl | 29.3 | 31.9 | xh | 6.4 | 28.9 |
| fr | 53.2 | 56.8 | lb | 24.2 | 1.1 | ps | 3.4 | 22.0 | yo | 3.4 | 4.2 |
| ga | 26.9 | 3.7 | lg | 3.6 | 12.5 | pt | 54.6 | 55.4 | zh | 30.7 | 27.1 |
| gl | 36.3 | 38.2 | ln | 5.8 | 26.0 | ro | 44.4 | 48.2 | zu | 6.6 | 32.2 |

Table 13: Comparison between ChatGPT and LegoMT2. Both in the En→X and X→En direction, ChatGPT falls behind LegoMT2 even with eight-shot.

| Model | X→En | En→X | AVG. |
|---|---|---|---|
| ChatGPT zero-shot | 27.9 | 23.9 | 25.9 |
| ChatGPT eight-shot | 31.9 | 24.7 | 28.3 |
| LegoMT2 | **38.3** | **31.6** | **35.0** |

The detailed results are shown in the table below Table 13. For some language pairs, the performance of ChatGPT is better than that of LegoMT2, such as En→Zh, where ChatGPT scores 30.7 versus LegoMT2 's 27.1. However, for the vast majority of language pairs, LegoMT2 has an absolute advantage. On average, ChatGPT lags behind LegoMT2 in both the En→X and X→En directions by more than 6 points.

**Comparison between ChatGPT with LegoMT2** A comparative analysis between ChatGPT (GPT 3.5) and LegoMT2 on 100 samples in *Flores-101*, as shown in Table 13, reveals that in zero-shot and eight-shot performance, ChatGPT lags behind LegoMT2 in the En→X and X→En direction more than 6 points. The prompts utilized for ChatGPT are "You are a helpful assistant that

translates {*SOURCE_LANG*} to {*TARGET_LANG*}." for the system and "Translate the following {*SOURCE_LANG*} text to {*TARGET_LANG*}: {*SOURCE_TEXT*}." for the user.

