# OpenReview forum: "LegoMT2: Non-Blocking Federated Learning for Massive Multilingual Machine Translation"
_ICLR.cc/2024/Conference — Submitted to ICLR 2024_

### Official Review · Reviewer_x6EA · 2023-10-21

**Soundness:** 3 good
**Presentation:** 3 good
**Contribution:** 2 fair
**Rating:** 3
**Confidence:** 5

**Summary:**

This paper proposes an efficient approach with a tailored model architecture for massive multilingual neural machine translation. LegoMT2
organizes 435 languages into 8 language-centric groups and attributes one local encoder-decoder for each group and a global encoder-decoder for all languages. LegoMT2 then trains each local and global encoder-decoder on a group-dedicated set of clients through asynchronous updating of parameters.

**Strengths:**

- federated learning used in MNMT to solve the parameter interference problem is somewhat novel
- This paper is well-written, and experiments show their improvements over baselines.

**Weaknesses:**

- The authors should present the key features of the traditional federated learning methods in the related works. The authors claim an efficient approach with a tailored model architecture for massive multilingual neural machine translation. What are the key attributes of the tailored model? In other words, what is the key difference between the federated learning used in this paper compared to the traditional federated method?
- The experimental results are somewhat less convincing. Actually, the model size of the model should be viewed as 10.4B rather than 1.6B. And the final model used in inference is the averaged version of the 8 local models. Therefore, the model should be compared to the same-size finetuned model.
- Why the model is finetuned from the pre-trained model? Why not training from scratch?

**Questions:**

- See above

---

> ### Author Response · Authors · 2023-11-22
>
> Thanks for the valuable comments and positive feedback on the novelty and effectiveness of our work. Please find our point-to-point responses below, which we hope will address any questions or concerns you may have.
>
> >**Question 1: What is the key difference between the federated learning used in this paper compared to the traditional federated method?**
>
> The key difference between the traditional federated method and our method is stated in Appendix A.  The main difference includes three aspects:  the data construction, the model architecture, and the communication method.
>
> |  | Traditional Federated Method | LegoMT2 |
> | --- | --- | --- |
> | Data Construction | Domain [1] | Language-centric grouping |
> | Model Architecture | Centralized  | Detachable |
> | Communication Method* | Synchronous | Asynchronous |
>
>
> One key factor in recent advanced FL algorithms is communication compression [2,3,4,5,6,7]. The traditional implementation of FL requires that each client send a full model (or a full model update) back to the server in each round. For large models, this step is likely to be the bottleneck of FL for a few reasons.
>
> **Reason 1:** internet speeds are usually faster for downloading than uploading. For example, Xfinity offers 125Mbps for downloads and only 15Mbps for uploads.
>
> **Reason 2:**  The size of the model is quite large. Therefore, there are numerous strategies available to compress these models or to reduce the bandwidth required for downloading the current model.
>
> **Reason 3:** Different clients may experience a considerable amount of idle time while waiting for the server to collect all models and generate a new global model.
>
> [1] Training mixed domain translation models via federated learning
> [2] Federated learning with compression: Unified analysis and sharp guarantees.
> [3] Federated learning: Strategies for improving communication efficiency
> [4] Communication-efficient adaptive federated learning
> [5] AutoFedNLP: An efficient FedNLP framework
> [6] FS-Real: Towards Real-World Cross-Device Federated Learning
> [7] FedED: Federated Learning via Ensemble Distillation for Medical Relation Extraction
>
> >**Question 2-1:  The final model used in inference is the averaged version of the 8 local models.**
>
> There seems to be a misunderstanding here: during inference (the results obtained in Table 1), we utilize only a single flow (mix-flow, 1.6B), not the averaged version of the 8 local models.
>
> >**Question 2-2:  The model size of the model should be viewed as 10.4B rather than 1.6B. Therefore, the model should be compared to the same-size finetuned model.**
> Yes,  we agree with your point. A fair comparing base model would be training 10.4B model and distilling into 1.6B model.
>
> **1) another valid comparison**
>
> We’ve given careful consideration to your suggestion and believe that, in addition to the model of the same size already listed in the table, there is indeed another model that could serve as a valid comparison:
>
> + LegoMT2: In training, the model of the whole system is 10.4B; but in inference, each flow can work independently, and the model size is only 1.6B.
>
> + Valid-Comparion:  We train a 10.4B model with centralized method, and then distilled this model to 1.6B.
>
> **2) It can be anticipated that this method will lag behind LegoMT2 in terms of both speed and performance.**
>
> + Time: As evident from Table 3, training a 10.4B model is significantly slower, specifically 16.2 times slower, than LegoMT2. Furthermore, this pipeline also necessitates the distillation of the 10.4B model to a 1.6B model. This process requires the inference of the 10.4B model to obtain the output of each sample, which significantly increases the time requirement.
>
> |  | Lego-MT2 | Valid-Comparison |
> | --- | --- | --- |
> | Training Speed | 16.2$\times$ | 1.0$\times$ |
> | distillation | - | Inference 10.4 to get the output of each sample |
>
> + Performance:  The 54.5B model’s average performance surpasses LegoMT-2 by one point. Therefore, the upper limit of the 10.4B model has been established, and it’s important to note that the distillation process inevitably leads to performance losses.
>
> Hence, when compared to the solution of training a 10.4B model and then distilling it, LegoMT2 proves to be both faster and more efficient.

---

> > ### Author Response · Authors · 2023-11-22
> >
> > >**Question 3: Why is the model finetuned from the pre-trained model? Why not training from scratch?**
> >
> > As the primary objective of LegoMT2 is to construct a system that supports 435 languages, it’s noteworthy that more than 300 of these are low-resource languages.
> >
> > However, controlling the quality of low resources is not easy, as shown in Table 2 of the  NLLB paper[1]:
> > |   |  |
> > |-------------------------------------------|-----|
> > | # of Languages requiring Re-translation   | 10  |
> > | Avg # of Re-translations                  | 1   |
> > | Max # of Re-translations                  | 2   |
> > | Avg # of Days to Translate 1 language     | 42  |
> > | Avg # of Days to align                    | 28  |
> > | Avg # of Days for 1 language              | 119 |
> > | Shortest Turnaround (days) for 1 language | 70  |
> > | Longest Turnaround (days) for 1 language  | 287 |
> > |   |  |
> >
> > By leveraging the existing model foundation, we can avoid redundancy in the process and devote more time to focusing on the newly added 200+ languages.
> >
> > [1] No Language Left Behind: Scaling Human-Centered Machine Translation

---

> > > ### Comment · Reviewer_x6EA · 2023-11-23
> > > **Response to the rebuttar**
> > >
> > > Hi, authors,
> > >    Thanks for your rebuttal. I have read your rebuttals to all reviewers and I re-read your paper carefully.
> > >    1  Actually, I have reviewed this paper two times (the last review in NIPS) and I noticed the improvement of your paper. However, I still cannot find some shining scientific points in your paper.
> > >    2 Can I say that you just apply the FL into multi-lingual NMT with little modification? As your rebuttal, it seems that little improvement have you made about FL you used in this paper.  Is there any detailed design for NMT.
> > >   3 The fair comparison is very important for a scientific research paper. We cannot just see the final report score, and more importantly, we need to care about why the improvement is achieved.
> > >   4 My question "why not train from scratch" shows my concern that your method may rely on a pre-trained NMT model.

---

> > > > ### Author Response · Authors · 2023-11-23
> > > >
> > > > Thanks for your response. We are pleased to have the opportunity to address your questions further.
> > > >
> > > > >**Can I say that you just apply the FL into multi-lingual NMT with little modification? As your rebuttal, it seems that little improvement have you made about FL you used in this paper. Is there any detailed design for NMT.**
> > > >
> > > > This work indeed applies FL to massive multilingual NMT pre-training tasks, which is a new training framework for massive multilingual machine translation pre-training tasks.  However, this is not an objective that is inherently apparent or readily achievable.
> > > > +  An available large-scale non-iid dataset is not easy to build with the traditional method
> > > > +  A large amount of computational and communication resources are required for pre-training a large NLP model
> > > > +  There is only one pre-training with the FL model, FLBERT[4], which trains with privacy data. Moreover, to reduce the communication requirements,  it only trains and updates the embedding layers.
> > > >
> > > > Below is a comparison between LegoMT2 and Traditional FL.
> > > >
> > > > **1)  data: The key challenge in FL: how to build the non-iid distribution**
> > > >
> > > > | Traditional FL | LegoMT2 |
> > > > | --- | --- |
> > > > | Method 1:  heuristics sampling [1] | Contribution： language-centric grouping  |
> > > > | Disadvantage of Method 1: such heuristics might fall short of replicating the complexity of natural heterogeneity found in real-world datasets[1], such as digital histopathology [2] | Advantage1:  Each dataset in the client is a real-world dataset, without heuristics |
> > > > | Method 2:  multilingual domain [3] | Advantage 2: Easy to obtain and plentiful |
> > > > | Disadvantages of Method 2: Domain restrictions, limited quantity |  |
> > > >
> > > >
> > > > **2)  model structure: The key challenge in the model is how to effectively train.**
> > > > | Traditional FL | LegoMT2 |
> > > > | --- | --- |
> > > > | centralized model architecture, only training with one global model | Contribution: detachable multi-way architecture |
> > > > | Method 1：Full model  | Advantage 1:  Applicable to pre-training setting |
> > > > | Disadvantage of Method 1:  constrained by network communication  | Advantage 2: Training with global model + local (designed for language-specific) model |
> > > > | Method 2: Only update the important parameters |  |
> > > > | Disadvantage of Method 2: hard to precisely identify the important parameters |  |
> > > > | Method 3: parameter efficient-tuning |  |
> > > > | Disadvantage of Method 3: Not applicable to pre-training |  |
> > > >
> > > > **3)  learning algorithm**:
> > > > | Traditional FL | LegoMT2 |
> > > > | --- | --- |
> > > > | Synchronous  | Asynchronous |
> > > > | Disadvantage 1: Large models need more uploading and downloading time. | Advantage: accelerated system training |
> > > > |  Disadvantage 2: Different clients may experience a considerable amount of idle time while waiting for the server to collect all models and generate a new global model. |  |
> > > >
> > > >
> > > > [1] FLamby: Datasets and Benchmarks for Cross-Silo Federated Learning in Realistic Healthcare Settings
> > > >
> > > > [2] Breast cancer histopathology image analysis: A review.
> > > >
> > > > [3] Training mixed domain translation models via federated learning
> > > >
> > > > [4] FedBERT: When Federated Learning Meets Pre-training

---

> ### Author Response · Authors · 2023-11-23
>
> >**The fair comparison is very important for a scientific research paper. We cannot just see the final report score, and more importantly, we need to care about why the improvement is achieved.**
>
> 1. We didn't merely report the final score, but also carried out extensive analysis experiments focusing on the key design elements of this training framework.
> + Table 4: model architecture analysis
> + Table 5: the language group strategy analysis
> + Figure 2: the effect of asynchronous learning algorithm on system performance.
> + Figure 3: the effect of hyperparameters in the training algorithm
>
> 2. Extensive experiments, coupled with multi-faced evaluations, have underscored the efficacy of the framework.
> + effectively supports translation between over 400 languages
> + evaluation of existing benchmark (Table 1 & Table 2)
> + human evaluation (Page 7-page 8 & Appendix D)
> + Compare with ChatGPT (Page 20)
>
> 3. This task involves pre-training, which inherently requires substantial computational and time resources.
> + It’s not that we are unwilling to make a fair comparison, but rather, we are unable to do so due to these constraints.
> |  | Lego-MT2 | Valid-Comparison |
> | --- | --- | --- |
> | Training Speed | 16.2$\times$ | 1.0$\times$ |
> | Training Time | 15 days on 64 A100 GPUs  | **240 days  on 64 A100 GPUs**  |
> | distillation | - | Inference 10.4 to get the output of each sample |
> | Overall Time | 15 days |  **240+ days** |
>
> + We can expect the performance of the solution (training a 10.4B model and then distilling to 1.6B) to be worse than LegoMT2.
> + The upper limit of the 10.4B model has been established. The 54.5B model’s average performance surpasses LegoMT-2 by one point. Therefore
> + It’s important to note that the distillation process inevitably leads to performance losses.
>
> Hence, when compared to the solution of training a 10.4B model and then distilling it, LegoMT2 proves to be both faster and more efficient.
>
> >**My question "why not train from scratch" shows my concern that your method may rely on a pre-trained NMT model.**
>
> 1) Utilizing NLLB-200-1.3B as an initial step is important, but it is not enough for handling 400+ languages.
>
> + Fine-tuning NLLB-200-1.3B directly on the data of over 400 languages actually results in a decrease in model performance, as shown in Table 1 of the paper.
> + One of our contributions to utilizing NLLB includes effort in extending the vocabulary from 200 languages to 435 languages. The tokens for an additional 235 languages are initiated randomly, resulting 0.3B extra parameters compared to NLLB. In this way, we can best utilize the pre-trained parameters in NLLB while largely expanding the language support in LegoMT2, in a cost-effective way.
>
>  2) The success of NLLB-200-1.3B is not trivial.   As shown below (which is from the NLLB paper[5]),  they devote considerable energy to optimizing language performance. On average, enhancing the performance of a language takes 119 days, with the most time-consuming language requiring as long as 287 days for optimization.
> |  |  |
> | --- | --- |
> | # of Languages requiring Re-translation | 10 |
> | Avg # of Re-translations | 1 |
> | Max # of Re-translations | 2 |
> | Avg # of Days to Translate 1 language | 42 |
> | Avg # of Days to align | 28 |
> | **Avg # of Days for 1 language** | **119** |
> | Shortest Turnaround (days) for 1 language | 70 |
> | **Longest Turnaround (days) for 1 language** | **287** |
> |  |  |
>
> [5] No Language Left Behind: Scaling Human-Centered Machine Translation
>
>  3) It is too costly to try full training of LegoMT2 without NLLB initialization.
>
> + Plan 1: Spending 287 days to replicate the performance of NLLB, and then adding more than 200+ languages does not seem to be an efficient solution.
> + Plan 2: Training each language from scratch and optimizing it, which takes over 200 days, is also costly.
>
> Considering limited computation resources, we want to utilize them to conduct more impactful research.

---

### Official Review · Reviewer_s6KB · 2023-11-01

**Soundness:** 4 excellent
**Presentation:** 3 good
**Contribution:** 4 excellent
**Rating:** 6
**Confidence:** 3

**Summary:**

To train a single model for massive languages is known for a challenging problem. This paper tackles the problem of how to efficiently train a neural machine translation for massive multilingual languages and proposed LegoMT2 that consists of local encoder-decoder models for language groups and a global encoder-decoder for all languages, where 435 languages are grouped into 8 language-centric category. The experimental results show the training efficiency and translation accuracy improvement, achieving 16.2x faster than the distributed training method for the same-size NLLLB and improving the translation accuracy by 2.2 BLEU on Flores-101 dataset averagely.

**Strengths:**

- The idea of asynchronous model parameter update that are language-group dependent is straightforward. Extensive experiments show that the proposed approach yields improvements in translation accuracy across languages. The proposed approach also helps the multi-way model to get trained faster.

**Weaknesses:**

- Extensive experimental results and analyses are not fit in 9 pages. There are some description overlaps in Section 1 and 3 so the authors can move the contents from Appendix to the main pages.

**Questions:**

- Reg Section 3.3; how helpful is the parameter initialization with NLLB-200-1.3B? Have you ever looked into this effect, without having the NLLB initialization?
- Have you ever tried with different language grouping?
- Why do you think Dec-Flows is better in the low-resource language groups?

---

> ### Author Response · Authors · 2023-11-22
>
> Thanks for your helpful suggestions. We will make the necessary changes.
>
> >**Weakness: Extensive experimental results and analyses are not fit in 9 pages. There are some description overlaps in Section 1 and 3 so the authors can move the contents from Appendix to the main pages.**
>
> Thanks for your suggestion! We will move the analysis of the difference between of traditional Federated Learning and comparison with LLM from the Appendix to the main part.
>
> >**Question 1: Reg Section 3.3; how helpful is the parameter initialization with NLLB-200-1.3B? Have you ever looked into this effect, without having the NLLB initialization?**
>
> We have a reasonable belief that NLLB initialization is important. However, it is too costly to try full training of LegoMT2 without NLLB initialization. Therefore, we did not conduct alternative experiments. According to NLLB paper[1] (Table 2), it would take over 200 days to replicate the alignment process of NLLB  to extend from NLLB's 200 languages to LegoMT2's 435 languages.
>
> |   | |
> |-------------------------------------------|-----|
> | # of Languages requiring Re-translation   | 10  |
> | Avg # of Re-translations                  | 1   |
> | Max # of Re-translations                  | 2   |
> | Avg # of Days to Translate 1 language     | 42  |
> | Avg # of Days to align                    | 28  |
> | Avg # of Days for 1 language              | 119 |
> | Shortest Turnaround (days) for 1 language | 70  |
> | Longest Turnaround (days) for 1 language  | 287 |
> |   | |
>
>
> [1] No Language Left Behind: Scaling Human-Centered Machine Translation
>
>
>
> Nevertheless, we would like to underscore that NLLB-200-1.3B only supports 200 languages. Our contribution to utilizing NLLB includes effort in extending the vocabulary from 200 languages to 435 languages. The tokens for an additional 235 languages are initiated randomly, resulting 0.3B extra parameters compared to NLLB. In this way, we can best utilize the pre-trained parameters in NLLB while largely expanding the language support in LegoMT2, in a cost-effective way.

---

> > ### Author Response · Authors · 2023-11-22
> >
> > >**Question 2: Have you ever tried with different language grouping?**
> >
> > We conducted an additional experiment with a different language grouping and found further increasing the group number can further improve system performance.
> >
> > The language information:
> >
> > | **new language group information = {**                                                                                                                                                                         |
> > |----------------------------------------------------------------------------------------------------------------------------------------------------------------------------------------------------------------|
> > |     "family_1": ['fr', 'es', 'en'],                                                                                                                                                                            |
> > |     "family_2": ['nl', 'tr', 'pl', 'it', 'de', 'pt'],                                                                                                                                                          |
> > |     "family_3": ['bg', 'ar', 'ru', 'fa', 'el', 'hu', 'ro', 'cs'],                                                                                                                                              |
> > |     "family_4": ['sk', 'da', 'uk', 'sl', 'he', 'fi', 'id', 'sv', 'vi'],                                                                                                                                        |
> > |     "family_5": ['ko', 'sq', 'hr', 'mk', 'sr', 'zh', 'no', 'bs', 'hi', 'ja', "zhtrad", 'zhtw', 'zhyue'],                                                                                                       |
> > |     "family_6": ['et', 'lt', 'lv', 'ca', 'bn', 'ms', 'th', 'gl', 'is', 'ta', 'sw', 'mt', 'eu', 'eo', 'tl', 'ml', 'af', 'si', 'ur', 'be', 'ne', 'mr', 'nb', 'ze', 'ka', 'sh'],                                  |
> > |     "family_7": ['az', 'ga', 'te', 'hy', 'mg', 'kk', 'cy', 'lb', 'ht', 'jv', 'ast', 'am', 'tg', 'km', 'mn', 'su', 'my', 'so', 'oc', 'xh', 'ha', 'ky', 'ps', 'pa', 'gu', 'ku', 'nds'],                          |
> > |     "family_8": ['fy', 'la', 'nn', 'kn', 'mi', 'yo', 'br', 'ig', 'sn', 'ny', 'hne', 'ceb', 'zu', 'uz', 'lo', 'cx', 'sd', 'wa', 'as', 'mai', 'se', 'pes', 'dje', 'pck', 'plt', 'crp'],                          |
> > |     "family_9": ['kek', 'tt', 'rw', 'fil', 'csb', 'yi', 'or', 'wo', 'ug', 'ilo', 'gd', 'ss', 'dv', 'hsb', 'crh', 'gv', 'tn', 'kab', 'an', 'nso', 'ba', 'kbh', 'syr', 'cop', 'chr', 'rom', 'quw'],              |
> > |     "family_10": ['cjp', 'dop', 'shi', 'ee', 'nhg', 'amu', 'wal', 'jak', 'usp', 'cni', 'li', 'ppk', 'quc',  'mam', 'hbs', 'ch', 'chq', 'cak', 'djk', 'cb', 'gbi', 'ff', 'fur', 'bsn', 'ojb', 'agr', 'dik'],    |
> > |     "family_11": ['jiv', 'ake', 'acu', 'lg', 'ia', 'tk', 'fo', 'om', 'ber', 'ln', 'tmh', 'pot', 'dz', 'ang',  'mhr', 'io', 'kg', 'bo', 'ns', 'ti', 'sa', 'kw', 'frp', 'arz', 'ckb', 'bar', 'wae'],             |
> > |     "family_12": ['ks', 'os', 'ace', 'sc', 'jbo', 'ts', 'arq', 'gom', 'szl', 'sco', 'ay', 'shn', 'zht', 'toki',  'zhs', 'mwl', 'kl', 'qu', 'zza', 'vec', 'ltg', 'aa', 'cv', 'ab', 'tlh', 'shs'],               |
> > |     "family_13": ['ie', 'tmp', 'bem', 'inh', 'pap', 'cmn', 'prs', 'lfn', 'ak', 'ce', 'lmo', 'tet', 'kr', 'wuu',  'st', 'lld', 'gn', 'rm', "nan", 'mus', 'mfe', 'miq', 've', 'scn', 'av', 'dsb', 'rn'],         |
> > |     "family_14": ['azb', 'pms', 'iu', 'bi', 'ase', 'bm', 'co', 'gr', 'kj', 'pmy', 'ry', 'lad', 'vo', 'efi',  'xal', 'fj', 'gos', 'nap', 'rup', 'ksh', 'tc', 'zsm', 'bh', 'nus', 'fuv', 'din', 'pi'],           |
> > |     "family_15": ['bal', 'cbk', 'brx', 'hil', 'cu', 'bua', 'dtp', 'pam', 'qd', 'hai', 'mos', 'cnh', 'avk',  'sml', 'lij', 'bug', 'kzj', 'trv', 'bnt', 'swg', 'to', 'tzl', 'zam', 'que', 'orv', 'kha',  'bho'], |
> > | }                                                                                                                                                                                                              |

---

> > > ### Author Response · Authors · 2023-11-22
> > >
> > > We allocated a tenth of the data for training the two grouping methods and subsequently assessed their performance on the Flores dataset. The method outlined in the paper is labeled as ‘old’. The results tabulated, such as ‘en-F1’, denote that English is the source language, while ‘F1’ encompasses all languages within Family-1. The findings suggest that a more sophisticated grouping design can further augment system performance.
> > >
> > > | **en-x** | **F1** | **F2** | **F3** | **F4** | **F5** | **F6** | **F7** | **F8** | **F9** | **F10** | **F11** |  | **x-en** | **F1** | **F2** | **F3** | **F4** | **F5** | **F6** | **F7** | **F8** | **F9** | **F10** | **F11** |
> > > |----------|--------|--------|--------|--------|--------|--------|--------|--------|--------|---------|---------|------|----------|--------|--------|--------|--------|--------|--------|--------|--------|--------|---------|---------|
> > > | old      | 41.8   | 36.7   | 35.0   | 37.5   | 30.0   | 31.7   | 26.8   | 25.7   | 19.4   | 0.8     | 12.1    |      | old      | 39.6   | 40.2   | 40.1   | 42.6   | 37.6   | 39.8   | 37.4   | 34.1   | 29.0   | 8.4     | 29.6    |
> > > | new      | 42.8   | 38.2   | 36.4   | 39.3   | 32.5   | 32.8   | 24.8   | 23.4   | 19.8   | 2.0     | 18.8    |      | new      | 40.6   | 41.1   | 41.6   | 44.0   | 39.2   | 40.7   | 38.1   | 32.9   | 29.0   | 10.7    | 30.2    |
> > > | **pt-x** | **F1** | **F2** | **F3** | **F4** | **F5** | **F6** | **F7** | **F8** | **F9** | **F10** | **F11** |  | **x-pt** | **F1** | **F2** | **F3** | **F4** | **F5** | **F6** | **F7** | **F8** | **F9** | **F10** | **F11** |
> > > | old      | 41.7   | 28.5   | 29.2   | 30.6   | 25.0   | 26.6   | 22.9   | 22.3   | 17.0   | 0.6     | 10.7    |      | old      | 41.0   | 32.9   | 34.3   | 35.0   | 31.4   | 32.4   | 30.4   | 26.8   | 23.6   | 7.1     | 23.0    |
> > > | new      | 42.5   | 29.9   | 31.0   | 32.4   | 26.5   | 25.7   | 22.0   | 20.4   | 16.2   | 2.0     | 14.0    |      | new      | 41.8   | 34.4   | 36.3   | 36.9   | 33.4   | 32.4   | 30.8   | 26.3   | 23.9   | 8.9     | 23.4    |
> > > | **hu-x** | **F1** | **F2** | **F3** | **F4** | **F5** | **F6** | **F7** | **F8** | **F9** | **F10** | **F11** |   | **x-hu** | **F1** | **F2** | **F3** | **F4** | **F5** | **F6** | **F7** | **F8** | **F9** | **F10** | **F11** |
> > > | old      | 32.8   | 26.3   | 25.2   | 26.0   | 21.9   | 23.0   | 19.9   | 19.4   | 15.2   | 0.6     | 9.0     |      | old      | 25.7   | 23.1   | 23.0   | 23.7   | 21.5   | 21.5   | 19.8   | 17.4   | 15.2   | 4.4     | 15.0    |
> > > | new      | 34.3   | 28.3   | 27.3   | 28.4   | 23.8   | 23.8   | 20.4   | 19.6   | 14.8   | 1.5     | 12.4    |      | new      | 27.9   | 25.7   | 25.9   | 26.3   | 24.1   | 22.1   | 21.0   | 17.2   | 16.0   | 6.1     | 15.5    |
> > > | **da-x** | **F1** | **F2** | **F3** | **F4** | **F5** | **F6** | **F7** | **F8** | **F9** | **F10** | **F11** |   | **x-da** | **F1** | **F2** | **F3** | **F4** | **F5** | **F6** | **F7** | **F8** | **F9** | **F10** | **F11** |
> > > | old      | 40.1   | 30.7   | 28.9   | 30.6   | 25.2   | 26.2   | 22.6   | 22.0   | 16.8   | 0.7     | 10.6    |      | old      | 36.6   | 31.4   | 31.2   | 32.3   | 28.8   | 29.4   | 27.3   | 24.2   | 20.8   | 6.1     | 21.0    |
> > > | new      | 41.6   | 32.6   | 31.0   | 32.6   | 27.5   | 27.6   | 23.1   | 22.3   | 17.4   | 1.6     | 15.0    |      | new      | 37.6   | 33.1   | 33.3   | 34.3   | 31.1   | 30.0   | 28.1   | 23.8   | 21.4   | 7.4     | 21.8    |
> > > | **zh-x** | **F1** | **F2** | **F3** | **F4** | **F5** | **F6** | **F7** | **F8** | **F9** | **F10** | **F11** |  | **x-zh** | **F1** | **F2** | **F3** | **F4** | **F5** | **F6** | **F7** | **F8** | **F9** | **F10** | **F11** |
> > > | old      | 26.0   | 20.7   | 20.2   | 20.9   | 18.9   | 19.0   | 15.9   | 16.4   | 11.6   | 0.5     | 6.4     |      | old      | 18.8   | 17.1   | 17.5   | 17.7   | 17.7   | 16.4   | 15.0   | 13.6   | 12.1   | 4.2     | 11.4    |
> > > | new      | 28.2   | 23.0   | 22.7   | 23.5   | 21.2   | 20.3   | 17.6   | 16.8   | 11.4   | 0.8     | 11.7    |      | new      | 20.9   | 19.2   | 19.8   | 20.0   | 20.1   | 17.7   | 16.7   | 13.9   | 13.6   | 5.7     | 13.0    |
> > > | **lg-x** | **F1** | **F2** | **F3** | **F4** | **F5** | **F6** | **F7** | **F8** | **F9** | **F10** | **F11** |   | **x-lg** | **F1** | **F2** | **F3** | **F4** | **F5** | **F6** | **F7** | **F8** | **F9** | **F10** | **F11** |
> > > | old      | 20.3   | 14.8   | 14.4   | 15.2   | 12.7   | 13.6   | 11.6   | 12.5   | 7.2    | 0.3     | 4.8     |      | old      | 10.8   | 9.5    | 9.3    | 9.9    | 8.1    | 8.2    | 7.5    | 7.5    | 4.9    | 1.5     | 7.6     |
> > > | new      | 18.3   | 12.0   | 10.1   | 13.2   | 11.5   | 12.7   | 10.5   | 11.7   | 6.4    | 0.6     | 10.4    |      | new      | 7.6    | 7.0    | 6.5    | 7.4    | 6.3    | 6.9    | 6.4    | 5.4    | 5.7    | 3.3     | 7.4     |

---

### Official Review · Reviewer_fThx · 2023-11-03

**Soundness:** 2 fair
**Presentation:** 3 good
**Contribution:** 2 fair
**Rating:** 5
**Confidence:** 3

**Summary:**

The paper presents a novel approach called LegoMT2 for multilingual neural machine translation. It addresses the challenge of learning a single model for a large number of languages by organizing languages into groups and using a multi-way model that includes multiple encoder-decoders – each for a certain language group and another global encoder-decoder. LegoMT2 trains these encoder-decoder pairs on dedicated server clients using asynchronous updating of parameters.

**Strengths:**

The proposed LegoMT2 supports over 400 languages for machine translation with one single encoder-decoder model, doubling the number of NLLB while significantly faster in training.

**Weaknesses:**

The paper did not conduct specific verification experiments on parameter interference to demonstrate that the performance improvement of LegoMT2 over finetuned NLLB-200-1.3B indeed stems from the alleviation of parameter interference phenomena.

**Questions:**

1. Which of Single-FT or Single-FT + MoE in Table 3 is used for the experiments in Table 1 and Table 2? Have the translation performance of both been evaluated?
2. Have any other methods for MERGE operation of non-blocking federated learning, apart from simple averaging, been tried and evaluated?
3. How about LLMs for Multilingual Machine Translation？

---

> ### Author Response · Authors · 2023-11-22
>
> Thank you very much for your insightful feedback and suggestions. Please see our responses to each of your comments listed below.
>
> >**Weakness: The paper did not conduct specific verification experiments on parameter interference to demonstrate that the performance improvement of LegoMT2 over finetuned NLLB-200-1.3B indeed stems from the alleviation of parameter interference phenomena.**
>
> Parameter interference is a fundamental problem in multilingual machine translation.
>
> It refers to the competition between different languages for the limited parameters of a model when we hope to use a single model to handle all translation directions. This can result in good translation results for some languages, while the translation results for other languages may be less satisfactory.
>
> As illustrated in Table 1, directly tuning NLLB-200-1.6B yields results that are inferior to those of NLLB-200-1.3B. However, LegoMT2 effectively enhances performance across over 400 languages, which suggests that LegoMT2 successfully mitigates parameter interference.
>
> >**Question 1: Which of Single-FT or Single-FT + MoE in Table 3 is used for the experiments in Table 1 and Table 2? Have the translation performance of both been evaluated?**
>
> The Single-FT is used for experiments in Table 1 and Table 2. We evaluated a pretrained 12B model (M2M-100-12B) but did not fine-tune the 12B model, because the training 12B model is too slow (16x slower in Table 3). It would require 240 days on 64 A100 GPUs to complete the training, which is prohibitively costly.
>
> >**Question 2：Have any other methods for MERGE operation of non-blocking federated learning, apart from simple averaging, been tried and evaluated?**
>
> No, we do not try other methods for MEREG.
>
> 1)  At present, this MERGE operation is adequate for LegoMT2.  The existing MERGE operation, which involves replacing the global module on the client side with the average of the parameters obtained from the server, is the standard merge operation in federated learning.
>
> 2) Testing different merge operations is a very resource- and time-consuming operation.  While we could explore more variations, it’s important to note that LegoMT2 represents a substantial pre-training effort.  LegoMT2's training takes 15 days on  64 80G A100 GPUs.
>
> 3) We sincerely hope that you reconsider the value of our paper. In this paper, we proposed a novel training framework, LegoMT2,  designed for massive massively multilingual machine translation (MNMT) systems.
>
>  + We introduced an efficient training framework for MNMT that supports 435 languages, which is more than any other system.
>  + LegoMT2 partitions the model and data together and employs an efficient non-blocking algorithm to accelerate training.
>  + LegoMT2 is small but powerful, which works only with 1.6B parameters and achieves better results than other models at the same or larger size (only behind NLLB-54B which is 30x larger).
>
> >**Question 3： How about LLMs for Multilingual Machine Translation？**
>
> The comparison with ChatGPT is in the Appendix. Both in the En→X and X→En direction, ChatGPT falls behind LegoMT2 even with eight-shot.
>
> | Model              | X$\rightarrow$En | En$\rightarrow$X | AVG. |
> |--------------------|------------------|------------------|------|
> | ChatGPT zero-shot  | 27.9             | 23.9             | 25.9 |
> | ChatGPT eight-shot | 31.9             | 24.7             | 28.3 |
> | LegoMT2            | 38.3             | 31.6             | 35.0 |

---

### Meta-Review · Area_Chair_jsb9 · 2023-12-09

**Metareview:**

This paper proposes an efficient approach with a tailored model architecture for massive multilingual neural machine translation. LegoMT2 organizes 435 languages into 8 language-centric groups and attributes one local encoder-decoder for each group and a global encoder-decoder for all languages.

All reviewers admit the system is very useful. The main concern is the scientific novelty of this paper. In addition, the empirical part is not so convincing.

**Justification For Why Not Higher Score:**

The main concern is the scientific novelty of this paper. In addition, the empirical part is not so convincing.

**Justification For Why Not Lower Score:**

n/a

---

### Decision · Program_Chairs · 2024-01-16

Reject